# RankFeat: Rank-1 Feature Removal for Out-of-distribution Detection

**Yue Song**[1], **Nicu Sebe**[1], and **Wei Wang**[2]

[1]Department of Information Engineering and Computer Science, University of Trento, Italy
[2]Beijing Jiaotong University, China
`yue.song@unitn.it`

## Abstract

The task of out-of-distribution (OOD) detection is crucial for deploying machine learning models in real-world settings. In this paper, we observe that the singular value distributions of the in-distribution (ID) and OOD features are quite different: the OOD feature matrix tends to have a larger dominant singular value than the ID feature, and the class predictions of OOD samples are largely determined by it. This observation motivates us to propose `RankFeat`, a simple yet effective `post hoc` approach for OOD detection by removing the rank-1 matrix composed of the largest singular value and the associated singular vectors from the high-level feature. `RankFeat` achieves *state-of-the-art* performance and reduces the average false positive rate (FPR95) by 17.90% compared with the previous best method. Extensive ablation studies and comprehensive theoretical analyses are presented to support the empirical results.

## 1 Introduction

In the real-world applications of deep learning, understanding whether a test sample belongs to the same distribution of training data is critical to the safe deployment of machine learning models. The main challenge stems from the fact that current deep learning models can easily give over-confident predictions for out-of-distribution (OOD) data [43]. Recently a rich line of literature has emerged to address the challenge of OOD detection [60, 27, 3, 7, 52, 14, 11, 54, 66, 37, 13, 68, 10, 15, 18].

Previous OOD detection approaches either rely on the feature distance [38], activation abnormality [52], or gradient norm [27]. In this paper, we tackle the problem of OOD detection from another perspective: by analyzing the spectrum of the high-level feature matrices (*e.g.,* the output of Block 3 or Block 4 of a typical ResNet [19] model), we observe that the feature matrices have quite different singular value distributions for the in-distribution (ID) and OOD data (see Fig. 1(a)): *the OOD feature tends to have a much larger dominant singular value than the ID feature, whereas the magnitudes of the rest singular values are very similar.* This peculiar behavior motivates us to remove the rank-1 matrix composed of the dominant singular value and singular vectors from the feature. As displayed in Fig. 1(b), removing the rank-1 feature drastically perturbs the class prediction of OOD samples; a majority of predictions have been changed. On the contrary, most ID samples have consistent classification results before and after removing the subspace. *This phenomenon indicates that the over-confident prediction of OOD samples might be largely determined by the dominant singular value and the corresponding singular vectors.*

Based on this observation, we **assume** that the first singular value of OOD feature tends to be much larger than that of ID feature. The intuition behind is that the OOD feature corresponds to a larger

---

Wei Wang is the corresponding author.

36th Conference on Neural Information Processing Systems (NeurIPS 2022).

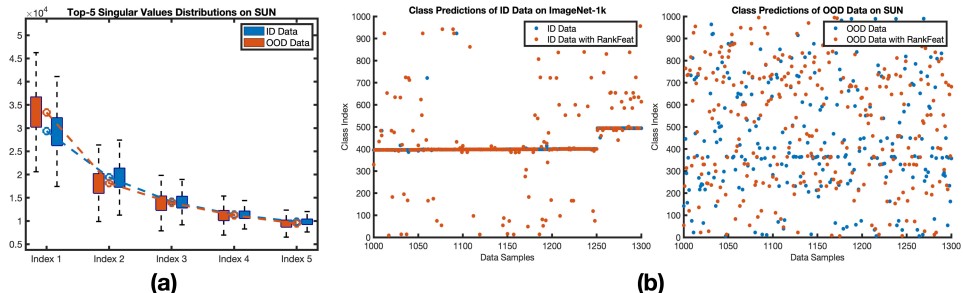

**(a)**            **(b)**

Figure 1: **(a)** The distribution of top-5 singular values for the ID and OOD features on ImageNet-1k and SUN. The OOD feature matrix tends to have a significantly larger dominant singular value. **(b)** After removing the rank-1 matrix composed by the dominant singular value and singular vectors, the class predictions of OOD data are severely perturbed, while those of ID data are moderately influenced. This observation indicates that the decisions of OOD data heavily depend on the dominant singular value and the corresponding singular vectors of the feature matrix. In light of this finding, we get motivated to propose `RankFeat` for OOD detection by removing the rank-1 matrix from the high-level feature. Both observations also hold for other OOD datasets.

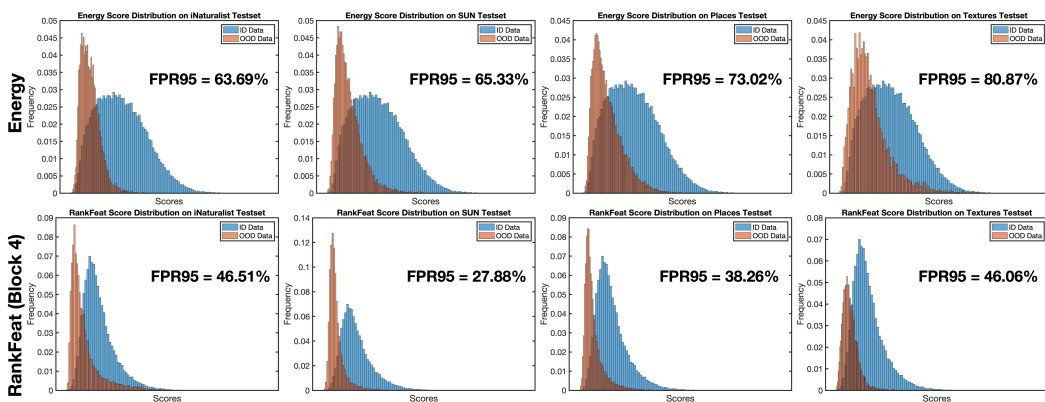

Figure 2: The score distributions of `Energy` [40] (top row) and our proposed `RankFeat` (bottom row) on four OOD datasets. Our method can better separate the ID and OOD data.

PCA explained variance ratio (being less informative), and the well-trained network weights might cause and amplify the difference (see Sec. E of the supplementary for the detailed illustration). Hence, we conjecture that leveraging this gap might help to better distinguish ID and OOD samples. To this end, we propose `RankFeat`, a simple but effective *post hoc* approach for OOD detection. `RankFeat` perturbs the high-level feature by removing its rank-1 matrix composed of the dominant singular value and vectors. Then the logits derived from the perturbed features are used to compute the OOD score function. By removing the rank-1 feature, the over-confidence of OOD samples is mitigated, and consequently the ID and OOD data can be better distinguished (see Fig. 2). Our `RankFeat` establishes the *state-of-the-art* performance on the large-scale ImageNet benchmark and a suite of widely used OOD datasets across different network depths and architectures. In particular, `RankFeat` outperforms the previous best method by **17.90%** in the average false positive rate (FPR95) and by **5.44%** in the area under curve (AUROC). Extensive ablation studies are performed to reveal important insights of `RankFeat`, and comprehensive theoretical analyses are conducted to explain the working mechanism. Code is publicly available via `https://github.com/KingJamesSong/RankFeat`.

The **key results and main contributions** are threefold:

- We propose `RankFeat`, a simple yet effective *post hoc* approach for OOD detection by removing the rank-1 matrix from the high-level feature. `RankFeat` achieves the *state-of-the-art* performance across benchmarks and models, reducing the average FPR95 by **17.90%** and improving the average AUROC by **5.44%** compared to the previous best method.

- We perform extensive ablation studies to illustrate the impact of (1) removing or keeping the rank-1 matrix, (2) removing the rank-n matrix (n>1), (3) applying our `RankFeat` at various network depths, (4) the number of iterations to iteratively derive the approximate rank-1 matrix for acceleration but without performance degradation, and (5) different fusion strategies to combine multi-scale features for further performance improvements.

- Comprehensive theoretical analyses are conducted to explain the working mechanism and to underpin the superior empirical results. We show that (1) removing the rank-1 matrix reduces the upper bound of OOD score more, (2) removing the rank-1 matrix makes the statistics of OOD feature closer to random matrices, and (3) both `RankFeat` and `ReAct` [52] work by optimizing the upper bound containing the largest singular value. `ReAct` [52] indirectly and manually clips the underlying term, while `RankFeat` directly subtracts it.

## 2 RankFeat: Rank-1 Feature Removal for OOD Detection

In this section, we introduce the background of OOD detection task and our proposed `RankFeat` that performs the OOD detection by removing the rank-1 matrix from the high-level feature.

**Preliminary: OOD detection.** The OOD detection is often formulated as a binary classification problem with the goal to distinguish between ID and OOD data. Let $f$ denote a model trained on samples from the ID data $\mathcal{D}_{in}$. For the unseen OOD data $\mathcal{D}_{out}$ at test time, OOD detection aims to define a decision function $\mathcal{G}(\cdot)$:

$$\mathcal{G}(\mathbf{x}) = \begin{cases} \text{in} & \mathcal{S}(\mathbf{x}) > \gamma, \\ \text{out} & \mathcal{S}(\mathbf{x}) < \gamma. \end{cases} \tag{1}$$

where $\mathbf{x}$ denotes the data encountered at the inference stage, $\mathcal{S}(\cdot)$ is the seeking scoring function, and $\gamma$ is a chosen threshold to make a large portion of ID data correctly classified (*e.g.,* $95\%$). The difficulty of OOD detection lies in designing an appropriate scoring function $\mathcal{S}(\cdot)$ such that the score distributions of ID and OOD data overlap as little as possible.

**RankFeat: rank-1 feature removal.** Consider the reshaped high-level feature map $\mathbf{X} \in \mathbb{R}^{C \times HW}$ of a deep network (the batch size is omitted for simplicity). Here 'high-level feature' denotes the feature map that carries rich semantics in the later layers of a network (*e.g.,* the output of Block 3 or Block 4 of a typical deep model like ResNet). Our `RankFeat` first performs the Singular Value Decomposition (SVD) on each individual feature matrix in the mini-batch to decompose the feature:

$$\mathbf{X} = \mathbf{U}\mathbf{S}\mathbf{V}^T \tag{2}$$

where $\mathbf{S} \in \mathbb{R}^{C \times HW}$ is the rectangular diagonal singular value matrix, and $\mathbf{U} \in \mathbb{R}^{C \times C}$ and $\mathbf{V} \in \mathbb{R}^{HW \times HW}$ are left and right orthogonal singular vector matrices, respectively. Then `RankFeat` removes the rank-1 matrix from the feature as:

$$\mathbf{X}' = \mathbf{X} - \mathbf{s}_1 \mathbf{u}_1 \mathbf{v}_1^T \tag{3}$$

where $\mathbf{s}_1$ is the largest singular value, and $\mathbf{u}_1$ and $\mathbf{v}_1$ are the corresponding left and right singular vectors, respectively. The perturbed feature is fed into the rest of the network to generate the logit predictions $\mathbf{y}'$. Finally, `RankFeat` computes the energy score of the logits for the input $\mathbf{x}$ as:

$$\texttt{RankFeat}(\mathbf{x}) = \log \sum \exp(\mathbf{y}') \tag{4}$$

By removing the rank-1 matrix composed by the dominant singular value $\mathbf{s}_1$, the over-confident predictions of OOD data are largely perturbed. In contrast, the decisions of ID data are mildly influenced. This could help to separate the ID and OOD data better in the logit space.

**Acceleration by Power Iteration.** Since `RankFeat` only involves the dominant singular value and vectors, there is no need to compute the full SVD of the feature matrix. Hence our method can be potentially accelerated by Power Iteration (PI). The PI algorithm is originally used to approximate the dominant eigenvector of a Hermitian matrix. With a slight modification, it can also be applied to general rectangular matrices. Given the feature $\mathbf{X}$, the modified PI takes the coupled iterative update:

$$\mathbf{v}_k = \frac{\mathbf{X}\mathbf{u}_k}{||\mathbf{X}\mathbf{u}_k||}, \ \mathbf{u}_{k+1} = \left(\frac{\mathbf{v}_k^T \mathbf{X}}{||\mathbf{v}_k^T \mathbf{X}||}\right)^T \tag{5}$$

where $\mathbf{u}_0$ and $\mathbf{v}_0$ are initialized with random orthogonal vectors and converge to the left and right singular vectors, respectively. After certain iterations, the dominant singular value is computed as $\mathbf{s}_1 = \mathbf{v}_k^T \mathbf{X} \mathbf{u}_k$. As will be illustrated in Sec. 4.3, the approximate solution yielded by PI achieves very competitive performance against the SVD but with much less time overhead.

**Combination of multi-scale features.** Our `RankFeat` works at various later depths of a model, *i.e.,* Block 3 and Block 4. Since intermediate features might focus on different semantic information, their decision cues are very likely to be different. It is thus natural to consider fusing the scores to leverage the distinguishable information of both features for further performance improvements. Let $\mathbf{y}'$ and $\mathbf{y}''$ denote the logit predictions of Block 3 and Block 4 features, respectively. `RankFeat` performs the fusion at the logit space and computes the score function as $\log \sum \exp((\mathbf{y}' + \mathbf{y}'')/2)$. Different fusion strategies are explored and discussed in Sec. 4.3.

# 3 Theoretical Analysis

In this section, we perform some theoretical analyses on `RankFeat` to support the empirical results. We start by proving that removing the rank-1 feature with a larger $\mathbf{s}_1$ would reduce the upper bound of `RankFeat` score more. Then based on Random Matrix Theory (RMT), we show that removing the rank-1 matrix makes the statistics of OOD features closer to random matrices. Finally, the theoretical connection of `ReAct` and our `RankFeat` is analyzed and discussed: both approaches work by optimizing the score upper bound determined by $\mathbf{s}_1$. `ReAct` manually uses a pre-defined threshold to clip the term with $\mathbf{s}_1$, whereas our `RankFeat` directly optimizes the bound by subtracting this term.

**Removing the rank-1 matrix with a larger $\mathbf{s}_1$ would reduce the upper bound of RankFeat more.** For our `RankFeat` score function, we can express its upper bound in an analytical form. Moreover, the upper bound analysis explicitly indicates that removing the rank-1 matrix with a larger first singular value would reduce the upper bound more. Specifically, we have the following proposition.

**Proposition 1.** *The upper bound of `RankFeat` score is defined as $\mathtt{RankFeat}(\mathbf{x}) < \frac{1}{HW}\left(\sum_{i=1}^{N} \mathbf{s}_i - \mathbf{s}_1\right)||\mathbf{W}||_\infty + ||\mathbf{b}||_\infty + \log(Q)$ where $Q$ denotes the number of categories, and $\mathbf{W}$ and $\mathbf{b}$ are the weight and bias of the last layer, respectively. A larger $\mathbf{s}_1$ would reduce the upper bound more.*

*Proof.* For the feature $\mathbf{X} \in \mathbb{R}^{C \times HW}$, its SVD $\mathbf{U}\mathbf{S}\mathbf{V}^T = \mathbf{X}$ can be expressed as the summation of rank-1 matrices $\mathbf{X} = \sum \mathbf{s}_i \mathbf{u}_i \mathbf{v}_i^T$. The feature perturbed by `RankFeat` can be computed as:

$$\mathbf{X}' = \mathbf{X} - \mathbf{s}_1 \mathbf{u}_1 \mathbf{v}_1^T = \sum_{i=2}^{N} \mathbf{s}_i \mathbf{u}_i \mathbf{v}_i^T \tag{6}$$

where $N$ denotes the shorter side of the matrix (usually $N = HW$). In most deep models [19, 20], usually the last feature map needs to pass a Global Average Pooling (GAP) layer to collapse the width and height dimensions. The GAP layer can be represented by a vector

$$\mathbf{m} = \frac{1}{HW} [1, 1, \ldots, 1]^T \tag{7}$$

The pooled feature map is calculated as $\mathbf{X}'\mathbf{m}$. Then the output logits are computed by the matrix-vector product with the classification head as:

$$\mathbf{y}' = \mathbf{W}\mathbf{X}'\mathbf{m} + \mathbf{b} = \sum_{i=2}^{N} (s_i \mathbf{W}\mathbf{u}_i \mathbf{v}_i^T \mathbf{m}) + \mathbf{b} \tag{8}$$

where $\mathbf{W} \in \mathbb{R}^{W \times C}$ denotes the weight matrix, $\mathbf{b} \in \mathbb{R}^{Q \times 1}$ represents the bias vector, and $\mathbf{y}' \in \mathbb{R}^{Q \times 1}$ is the output logits that correspond to the perturbed feature $\mathbf{X}'$. Our `RankFeat` score is computed as:

$$\mathtt{RankFeat}(\mathbf{x}) = \log \sum_{i=1}^{Q} \exp(\mathbf{y}_i') \tag{9}$$

where $\mathbf{x}$ is the input image, and $Q$ denotes the number of categories. Here we choose Energy [40] as the base function due to its theoretical alignment with the input probability density and its strong

empirical performance. Eq. (9) can be re-formulated by the `Log-Sum-Exp` trick

$$\log \sum_{i=1}^{Q} \exp(\mathbf{y}'_i) = \log \sum_{i=1}^{Q} \exp(\mathbf{y}'_i - \max(\mathbf{y}')) + \max(\mathbf{y}') \tag{10}$$

The above equation directly yields the tight bound as:

$$\max(\mathbf{y}') < \log \sum \exp(\mathbf{y}') < \max(\mathbf{y}') + \log(Q) \tag{11}$$

Since $\max(\mathbf{y}') \leq \max(|\mathbf{y}'|) = ||\mathbf{y}'||_\infty$, we have

$$\texttt{RankFeat}(\mathbf{x}) = \log \sum \exp(\mathbf{y}') < \max(\mathbf{y}') + \log(Q) \leq ||\mathbf{y}'||_\infty + \log(Q) \tag{12}$$

The vector norm has the property of triangular inequality, *i.e.,* $||\mathbf{a} + \mathbf{c}|| \leq ||\mathbf{a}|| + ||\mathbf{c}||$ holds for any vectors $\mathbf{a}$ and $\mathbf{c}$. Moreover, since both $\mathbf{u}$ and $\mathbf{v}$ are orthogonal vectors, we have the relation $||\mathbf{u}_i||_\infty \leq 1$ and $||\mathbf{v}_i||_\infty \leq 1$. Relying on these two properties, injecting eq. (8) into eq. (12) leads to

$$\texttt{RankFeat}(\mathbf{x}) < \sum_{i=2}^{N} \mathbf{s}_i ||\mathbf{W}\mathbf{u}_i\mathbf{v}_i^T\mathbf{m}||_\infty + ||\mathbf{b}||_\infty + \log(Q) \leq \sum_{i=2}^{N} \mathbf{s}_i ||\mathbf{W}\mathbf{m}||_\infty + ||\mathbf{b}||_\infty + \log(Q) \tag{13}$$

Since $\mathbf{m}$ is a scaled all-ones vector, we have $||\mathbf{W}\mathbf{m}||_\infty = ||\mathbf{W}||_\infty / HW$. The bound is simplified as:

$$\texttt{RankFeat}(\mathbf{x}) < \frac{1}{HW} \Big( \sum_{i=1}^{N} \mathbf{s}_i - \mathbf{s}_1 \Big) ||\mathbf{W}||_\infty + ||\mathbf{b}||_\infty + \log(Q) \tag{14}$$

As indicated above, removing a larger $\mathbf{s}_1$ would reduce the upper bound of `RankFeat` score more. □

**Remark:** Considering that OOD feature usually has a much larger $\mathbf{s}_1$ (see Fig. 1(a)), `RankFeat` would reduce the upper bound of OOD samples more.

*Notice that our bound analysis strives to improve the understanding of OOD methods from new perspectives instead of giving a strict guarantee of the score.* For example, the upper bound can be used to explain the shrinkage and skew of score distributions in Fig. 2. Subtracting $\mathbf{s}_1$ would largely reduce the numerical range of both ID and OOD scores, which could squeeze score distributions. Since the dominant singular value $\mathbf{s}_1$ contributes most to the score, removing $\mathbf{s}_1$ is likely to make many samples have similar scores. This would concentrate samples in a smaller region and further skew the distribution. Given that the OOD feature tends to have a much larger $\mathbf{s}_1$, this would have a greater impact on OOD data and skew the OOD score distribution more.

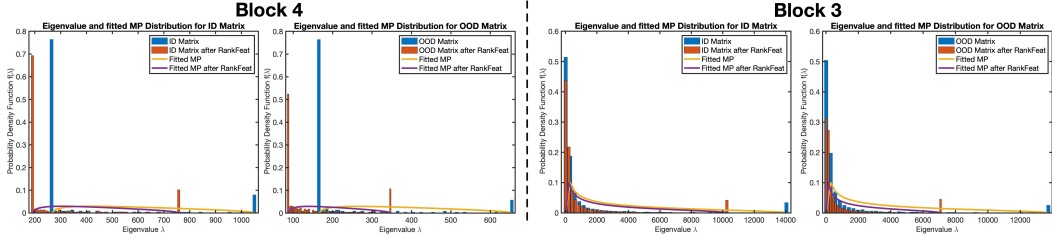

Figure 3: The exemplary eigenvalue distribution of ID/OOD feature and the fitted MP distribution. After the rank-1 matrix is removed, the lowest bin of OOD feature has a larger reduction and the middle bins gain some growth, making the ODD feature statistics closer to the MP distribution.

**Removing the rank-1 matrix is likely to make the statistics of OOD features closer to random matrices.** Now we turn to use RMT to analyze the statistics of OOD and ID feature matrices. For a random matrix of a given shape, the density of its eigenvalue asymptotically converges to the Manchenko-Pastur (MP) distribution [41, 50]. Formally, we have:

**Theorem 1** (Manchenko-Pastur Law [41, 50])**.** *Let $\mathbf{X}$ be a random matrix of shape $t \times n$ whose entries are random variables with $E(\mathbf{X}_{ij} = 0)$ and $E(\mathbf{X}_{ij}^2 = 1)$. Then the eigenvalues of the sample covariance $\mathbf{Y} = \frac{1}{n}\mathbf{X}\mathbf{X}^T$ converges to the probability density function: $\rho(\lambda) = \frac{t}{n} \frac{\sqrt{(\lambda_+ - \lambda)(\lambda - \lambda_-)}}{2\pi\lambda\sigma^2}$ for $\lambda \in [\lambda_-, \lambda_+]$ where $\lambda_- = \sigma^2(1 - \sqrt{\frac{n}{t}})^2$ and $\lambda_+ = \sigma^2(1 + \sqrt{\frac{n}{t}})^2$.*

This theorem implies the possibility to measure the statistical distance between ID/OOD features and random matrices. To this end, we randomly sample $1,000$ ID and OOD feature matrices and compute the KL divergence between the actual eigenvalue distribution and the fitted MP distribution.

Table 1: The KL divergence between ID/OOD feature and the fitted MP distribution. When the rank-1 feature is removed, the statistics of OOD matrix are closer to random matrices.

| Matrix Type | Block 4 | | Block 3 | |
|---|---|---|---|---|
| | ID | OOD | ID | OOD |
| Original feature matrix | 18.36 | 18.24 | 11.27 | 11.18 |
| Removing rank-1 matrix | 17.07 ($\downarrow$ 1.29) | **15.79 ($\downarrow$ 2.45)** | 9.84 ($\downarrow$ 1.45) | **8.71 ($\downarrow$ 2.47)** |

Fig. 3 and Table 1 present the exemplary eigenvalue distribution and the average evaluation results of Block 4 and Block 3 features, respectively. For the original feature, the OOD and ID feature matrices exhibit similar behaviors: the distances to the fitted MP distribution are roughly the same ($diff.\approx 0.1$). However, when the rank-1 matrix is removed, the OOD feature matrix has a much larger drop in the KL divergence. This indicates that removing the rank-1 matrix makes the statistics of OOD feature closer to random matrices, *i.e.,* the OOD feature is very likely to become less informative than the ID feature. The result partly explains the working mechanism of `RankFeat`: *by removing the feature matrix where OOD data might convey more information than ID data, the two types of distributions have a larger discrepancy and therefore can be better separated.*

**Connection with ReAct [52].** `ReAct` clips the activations at the penultimate layer of a model to distinguish ID and OOD samples. Given the feature $\mathbf{X}$ and the pooling layer $\mathbf{m}$, the perturbation can be defined as:

$$\min(\mathbf{X}\mathbf{m}, \tau) = \mathbf{X}\mathbf{m} - \max(\mathbf{X}\mathbf{m} - \tau, 0) \tag{15}$$

where $\tau$ is a pre-defined threshold. Their method shares some similarity with `RankFeat` formulation $\mathbf{X}\mathbf{m} - \mathbf{s}_1\mathbf{u}_1\mathbf{v}_1^T\mathbf{m}$. *Both approaches subtract from the feature a portion of information that is most likely to cause the over-confidence of OOD prediction.* `ReAct` selects the manually-defined threshold $\tau$ based on statistics of the whole ID set, while `RankFeat` generates the structured rank-1 matrix from the feature itself. Taking a step further, `ReAct` has the score inequality following eq. (12)

$$\texttt{ReAct}(\mathbf{x}) < ||\mathbf{W}\mathbf{X}\mathbf{m} - \mathbf{W}\max(\mathbf{X}\mathbf{m} - \tau, 0)||_\infty + ||\mathbf{b}||_\infty + \log(Q) \tag{16}$$

Since $\mathbf{X}$ is non-negative (output of `ReLU`), we have $\max(\mathbf{X}\mathbf{m}) \geq {}^{\max(\mathbf{X})}/_{HW}$. Exploiting the vector norm inequality $||\mathbf{X}||_\text{F} \geq ||\mathbf{X}||_2$ leads to the relation $\max(\mathbf{X}) \geq {}^{\mathbf{s}_1}/_{\sqrt{CHW}}$. Relying on this property, the above inequality can be re-formulated as:

$$\texttt{ReAct}(\mathbf{x}) < \frac{1}{HW}\sum_{i=1}^{N}\mathbf{s}_i||\mathbf{W}||_\infty - \boxed{\frac{1}{HW}\max(\frac{\mathbf{s}_1}{\sqrt{CHW}} - \tau, 0)||\mathbf{W}||_\infty} + ||\mathbf{b}||_\infty + \log(Q) \tag{17}$$

As indicated above, the upper bound of `ReAct` is also determined by the largest singular value $\mathbf{s}_1$. In contrast, the upper bound of our `RankFeat` can be expressed as:

$$\texttt{RankFeat}(\mathbf{x}) < \frac{1}{HW}\sum_{i=1}^{N}\mathbf{s}_i||\mathbf{W}||_\infty - \boxed{\frac{1}{HW}\mathbf{s}_1||\mathbf{W}||_\infty} + ||\mathbf{b}||_\infty + \log(Q) \tag{18}$$

The upper bounds of both methods resemble each other with the only different term boxed. *From this point of view, both methods distinguish the ID and OOD data by eliminating the impact of the term containing $\mathbf{s}_1$ in the upper bound.* `ReAct` optimizes it by clipping the term with a manually-defined threshold, which is indirect and might be sub-optimal. Moreover, the threshold selection requires statistics of the whole ID set. In contrast, our `RankFeat` does not require any extra data and directly subtracts this underlying term which is likely to cause the over-confidence of OOD samples.

## 4 Experimental Results

In this section, we first discuss the setup in Sec. 4.1, and then present the main experimental results on ImageNet-1k in Sec. 4.2, followed by the extensive ablation studies in Sec. 4.3.

## 4.1 Setup

**Datasets.** In line with [26, 52, 27], we mainly evaluate our method on the large-scale ImageNet-1k benchmark [6]. The large-scale dataset is more challenging than the traditional CIFAR benchmark [36] because the images are more realistic and diverse (*i.e.,* 1.28M images of 1, 000 classes). For the OOD datasets, we select four testsets from subsets of `iNaturalist` [58], `SUN` [63], `Places` [70], and `Textures` [5]. These datasets are crafted by [26] with non-overlapping categories from ImageNet-1k. Besides the experiment on the large-scale benchmark, we also validate the effectiveness of our approach on Species [24] and CIFAR [36] benchmark. (see Supplementary Material).

**Baselines.** We compare our method with 6 recent *post hoc* OOD detection methods, namely `MSP` [22], `ODIN` [39], `Energy` [40], `Mahalanobis` [38], `GradNorm` [27], and `ReAct` [52]. The detailed illustration and settings of these methods are kindly referred to Supplementary Material.

**Architectures.** In line with [27], the main evaluation is done using Google BiT-S model [35] pretrained on ImageNet-1k with ResNetv2-101 [20]. We also evaluate the performance on SqueezeNet [29], an alternative tiny architecture suitable for mobile devices and on T2T-ViT-24 [67], a tokens-to-tokens vision transformer that has impressive performance when trained from scratch.

*For the implementation details and evaluation metrics, please refer to Supplementary Material.*

Table 2: Main evaluation results on ResNetv2-101 [20]. All values are reported in percentages, and these *post hoc* methods are directly applied to the model pre-trained on ImageNet-1k [6]. The best three results are highlighted with **red**, **blue**, and **cyan**.

| Methods | iNaturalist | | SUN | | Places | | Textures | | Average | |
|---|---|---|---|---|---|---|---|---|---|---|
| | FPR95 ($\downarrow$) | AUROC ($\uparrow$) | FPR95 ($\downarrow$) | AUROC ($\uparrow$) | FPR95 ($\downarrow$) | AUROC ($\uparrow$) | FPR95 ($\downarrow$) | AUROC ($\uparrow$) | FPR95 ($\downarrow$) | AUROC ($\uparrow$) |
| MSP [22] | 63.69 | 87.59 | 79.89 | 78.34 | 81.44 | 76.76 | 82.73 | 74.45 | 76.96 | 79.29 |
| ODIN [39] | 62.69 | 89.36 | 71.67 | 83.92 | 76.27 | 80.67 | 81.31 | 76.30 | 72.99 | 82.56 |
| Energy [40] | 64.91 | 88.48 | 65.33 | 85.32 | 73.02 | 81.37 | 80.87 | 75.79 | 71.03 | 82.74 |
| Mahalanobis [38] | 96.34 | 46.33 | 88.43 | 65.20 | 89.75 | 64.46 | 52.23 | 72.10 | 81.69 | 62.02 |
| GradNorm [27] | 50.03 | 90.33 | 46.48 | 89.03 | 60.86 | 84.82 | 61.42 | 81.07 | 54.70 | 86.71 |
| ReAct [52] | 44.52 | 91.81 | 52.71 | 90.16 | 62.66 | 87.83 | 70.73 | 76.85 | 57.66 | 86.67 |
| **RankFeat (Block 4)** | 46.54 | 81.49 | 27.88 | 92.18 | 38.26 | 88.34 | 46.06 | 89.33 | 39.69 | 87.84 |
| **RankFeat (Block 3)** | 49.61 | 91.42 | 39.91 | 92.01 | 51.82 | 88.32 | 41.84 | 91.44 | 45.80 | 90.80 |
| **RankFeat (Block 3 + 4)** | 41.31 | 91.91 | 29.27 | 94.07 | 39.34 | 90.93 | 37.29 | 91.70 | 36.80 | 92.15 |

## 4.2 Results

**Main results.** Following [27], the main evaluation is conducted using Google BiT-S model [35] pretrained on ImageNet-1k with ResNetv2-101 architecture [20]. Table 2 compares the performance of all the *post hoc* methods. For both Block 3 and Block 4 features, our `RankFeat` achieves the best evaluation results across datasets and metrics. More specifically, `RankFeat` based on the Block 4 feature outperforms the second-best baseline by **15.01%** in the average FPR95, while the Block 3 feature-based `RankFeat` beats the second-best method by **4.09%** in the average AUROC. Their combination further surpasses other methods by **17.90%** in the average FPR95 and by **5.44%** in the average AUROC. The superior performances at various depths demonstrate the effectiveness and general applicability of `RankFeat`. The Block 3 feature has a higher AUROC but slightly falls behind the Block 4 feature in the FPR95, which can be considered a compromise between the two metrics.

**Our RankFeat is also effective on alternative CNN architectures.** Besides the experiment on ResNetv2 [20], we also evaluate our method on SqueezeNet [29], an alternative tiny network suitable for mobile devices and on-chip applications. This network is more challenging because the tiny network size makes the model prone to overfit the training data, which could increase the difficulty to distinguish between ID and OOD samples. Table 3 top presents the performance of all the methods. Collectively, the performances of `RankFeat` are very competitive at both depths, as well as the score fusion. Our `RankFeat` achieves the *state-of-the-art* performances, outperforming the second-best baseline by **14.22%** in FPR95 and by **7.48%** in AUROC.

**Our RankFeat also suits transformer-based architectures.** To further demonstrate the applicability of our method, we evaluate `RankFeat` on Tokens-to-Tokens Vision Transformer (T2T-ViT) [67], a popular transformer architecture that can achieve competitive performance against CNNs when trained from scratch. Similar to the CNN, `RankFeat` removes the rank-1 matrix from the final token

Table 3: The results on SqueezeNet [29] and T2T-ViT-24 [67]. All values are reported in percentages, and these *post hoc* methods are directly applied to the model pre-trained on ImageNet-1k [6]. For results on SqueezeNet [29], the best three results are highlighted with **red**, **blue**, and **cyan**.

| Model | Methods | iNaturalist | | SUN | | Places | | Textures | | Average | |
|---|---|---|---|---|---|---|---|---|---|---|---|
| | | FPR95 (↓) | AUROC (↑) | FPR95 (↓) | AUROC (↑) | FPR95 (↓) | AUROC (↑) | FPR95 (↓) | AUROC (↑) | FPR95 (↓) | AUROC (↑) |
| SqueezeNet [29] | MSP [22] | 89.83 | 65.41 | 83.03 | 72.25 | 87.27 | 67.00 | 94.61 | 41.84 | 88.84 | 61.63 |
| | ODIN [39] | 90.79 | 65.75 | 78.32 | 78.37 | 83.23 | 73.31 | 92.25 | 43.43 | 86.15 | 65.17 |
| | Energy [40] | 79.27 | 73.30 | 56.41 | 87.88 | 67.74 | 82.73 | 67.16 | 64.51 | 67.65 | 77.11 |
| | Mahalanobis [38] | 91.50 | 51.79 | 90.33 | 62.18 | 92.26 | 56.63 | 58.60 | 67.16 | 83.17 | 59.44 |
| | GradNorm [27] | 76.31 | 73.92 | 53.63 | 87.55 | 65.99 | 83.28 | 68.72 | 68.07 | 66.16 | 78.21 |
| | ReAct [52] | 76.78 | 68.56 | 87.57 | 66.37 | 88.80 | 66.20 | 51.05 | 76.57 | 76.05 | 69.43 |
| | **RankFeat (Block 4)** | 61.67 | 83.09 | 46.72 | 88.31 | 61.31 | 80.52 | 38.04 | 88.82 | 51.94 | 85.19 |
| | **RankFeat (Block 3)** | 71.04 | 81.50 | 49.18 | 90.43 | 62.94 | 85.82 | 50.14 | 79.32 | 58.33 | 84.28 |
| | **RankFeat (Block 3 + 4)** | 65.81 | 83.06 | 46.64 | 90.17 | 61.56 | 84.51 | 42.54 | 85.00 | 54.14 | 85.69 |
| T2T-ViT-24 [67] | MSP [22] | 48.92 | 88.95 | 61.77 | 81.37 | 69.54 | 80.03 | 62.91 | 82.31 | 60.79 | 83.17 |
| | ODIN [39] | 44.07 | 88.17 | 63.83 | 78.46 | 68.19 | 75.33 | 54.27 | 83.63 | 57.59 | 81.40 |
| | Energy [40] | 52.95 | 82.93 | 68.55 | 73.06 | 74.24 | 68.17 | 51.05 | 83.25 | 61.70 | 76.85 |
| | Mahalanobis [38] | 90.50 | 58.13 | 91.71 | 50.52 | 93.32 | 49.60 | 80.67 | 64.06 | 89.05 | 55.58 |
| | GradNorm [27] | 99.30 | 25.86 | 98.37 | 28.06 | 99.01 | 25.71 | 92.68 | 38.80 | 97.34 | 29.61 |
| | ReAct [52] | 52.17 | 89.51 | 65.23 | 81.03 | 68.93 | 78.20 | 52.54 | 85.46 | 59.72 | 83.55 |
| | **RankFeat** | 50.27 | 87.81 | 57.18 | 84.33 | 66.22 | 80.89 | 32.64 | 89.36 | 51.58 | 85.60 |

of T2T-ViT before the last normalization layer and the classification head. Table 3 bottom compares the performance on T2T-ViT-24. Our `RankFeat` outperforms the second-best method by **6.11%** in FPR95 and by **2.05%** in AUROC. Since the transformer models [9, 67] do not have increasing receptive fields like CNNs, we do not evaluate the performance at alternative network depths.

**Comparison against training-needed approaches.** Since our method is *post hoc*, we only compare it with other *post hoc* baselines. MOS [26] and KL Matching [24] are not taken into account because MOS needs extra training processes and KL Matching requires the labeled validation set to compute distributions for each class. Nonetheless, we note that our method can still hold an advantage against those approaches. Table 4 presents the average FPR95 and AUROC on the ImageNet-1k benchmark. Our RankFeat achieves the best performance without any extra training or validation set.

Table 4: Comparison against training-needed methods on ImageNet-1k based on ResNetv2-101 [20].

| Method | Post hoc? | Free of Validation Set? | FPR95 (↓) | AUROC (↑) |
|---|---|---|---|---|
| KL Matching [24] | ✓ | ✗ | 54.30 | 80.82 |
| MOS [26] | ✗ | ✓ | 39.97 | 90.11 |
| **RankFeat** | ✓ | ✓ | **36.80** | **92.15** |

## 4.3 Ablation Studies

In this subsection, we conduct several ablation studies based on Google-BiT-S ResNetv2-101 model. Unless explicitly specified, we apply `RankFeat` on the Block 4 feature by default.

Table 5: Ablation studies on keeping only the rank-1 matrix and removing the rank-n matrix.

| Baselines | iNaturalist | | SUN | | Places | | Textures | | Average | |
|---|---|---|---|---|---|---|---|---|---|---|
| | FPR95 (↓) | AUROC (↑) | FPR95 (↓) | AUROC (↑) | FPR95 (↓) | AUROC (↑) | FPR95 (↓) | AUROC (↑) | FPR95 (↓) | AUROC (↑) |
| GradNorm [27] | 50.03 | 90.33 | 46.48 | 89.03 | 60.86 | 84.82 | 61.42 | 81.07 | 54.70 | 86.71 |
| ReAct [52] | 44.52 | 91.81 | 52.71 | 90.16 | 62.66 | 87.83 | 70.73 | 76.85 | 57.66 | 86.67 |
| Keeping Only Rank-1 | 48.97 | 91.93 | 62.63 | 84.62 | 72.42 | 79.79 | 49.42 | 88.86 | 58.49 | 86.30 |
| Removing Rank-3 | 55.19 | 90.03 | 48.97 | 91.26 | 56.63 | 88.81 | 86.95 | 74.57 | 61.94 | 86.17 |
| Removing Rank-2 | 50.04 | 89.30 | 48.55 | 90.99 | 56.23 | 88.38 | 76.86 | 81.37 | 57.92 | 87.51 |
| **Removing Rank-1** | 46.54 | 81.49 | 27.88 | 92.18 | 38.26 | 88.34 | 46.06 | 89.33 | 39.69 | 87.84 |

**Removing the rank-1 matrix outperforms keeping only it.** Instead of removing the rank-1 matrix, another seemingly promising approach is keeping only the rank-1 matrix and abandoning the rest of the matrix. Table 5 presents the evaluation results of keeping only the rank-1 matrix. The performance falls behind that of removing the rank-1 feature by 18.8% in FPR95, which indicates that keeping only the rank-1 feature is inferior to removing it in distinguishing the two distributions. Nonetheless,

it is worth noting that even keeping only the rank-1 matrix achieves very competitive performance against previous best methods, such as `GradNorm` [27] and `ReAct` [52].

**Removing the rank-1 matrix outperforms removing the rank-n matrix (n>1).** We evaluate the impact of removing the matrix of a higher rank, *i.e.,* performing $\mathbf{X} - \sum_{i=1}^{n} \mathbf{s}_i \mathbf{u}_i \mathbf{v}_i^T$ where $n > 1$ for the high-level feature $\mathbf{X}$. Table 5 compares the performance of removing the rank-2 matrix and rank-3 matrix. When the rank of the removed matrix is increased, the average performance degrades accordingly. This demonstrates that removing the rank-1 matrix is the most effective approach to separate ID and OOD data. This result is coherent with the finding in Fig. 1(a): only the largest singular value of OOD data is significantly different from that of ID data. Therefore, removing the rank-1 matrix achieves the best performance.

Table 6: The ablation study on applying `RankFeat` to features at different network depths.

| Layer | iNaturalist | | SUN | | Places | | Textures | | Average | |
|---|---|---|---|---|---|---|---|---|---|---|
| | FPR95 ($\downarrow$) | AUROC ($\uparrow$) | FPR95 ($\downarrow$) | AUROC ($\uparrow$) | FPR95 ($\downarrow$) | AUROC ($\uparrow$) | FPR95 ($\downarrow$) | AUROC ($\uparrow$) | FPR95 ($\downarrow$) | AUROC ($\uparrow$) |
| Block 1 | 87.81 | 77.00 | 59.15 | 87.29 | 65.50 | 84.35 | 94.15 | 60.41 | 76.65 | 77.26 |
| Block 2 | 71.84 | 85.80 | 61.44 | 86.46 | 71.68 | 81.65 | 87.89 | 72.04 | 73.23 | 81.49 |
| **Block 3** | **49.61** | **91.42** | **39.91** | **92.01** | **51.82** | **88.32** | **41.84** | **91.44** | **45.80** | **90.80** |
| **Block 4** | **46.54** | 81.49 | **27.88** | **92.18** | **38.26** | **88.34** | 46.06 | 89.33 | **39.69** | 87.84 |

**Block 3 and Block 4 features are the most informative.** In addition to exploring the high-level features at Block 3 and Block 4, we also investigate the possibility of applying `RankFeat` to features at shallow network layers. As shown in Table 6, the performances of `RankFeat` at the Block 1 and Block 2 features are not comparable to those at deeper layers. This is mainly because the shallow low-level features do not embed as rich semantic information as the deep features. Consequently, removing the rank-1 matrix of shallow features would not help to separate the ID and OOD data.

Table 7: The approximate solution by PI yields competitive performance and costs much less time consumption. The test batch size is set as 16.

| Computation Technique | Processing Time Per Image (ms) | iNaturalist | | SUN | | Places | | Textures | | Average | |
|---|---|---|---|---|---|---|---|---|---|---|---|
| | | FPR95 ($\downarrow$) | AUROC ($\uparrow$) | FPR95 ($\downarrow$) | AUROC ($\uparrow$) | FPR95 ($\downarrow$) | AUROC ($\uparrow$) | FPR95 ($\downarrow$) | AUROC ($\uparrow$) | FPR95 ($\downarrow$) | AUROC ($\uparrow$) |
| GradNorm [27] | 80.01 | 50.03 | 90.33 | 46.48 | 89.03 | 60.86 | 84.82 | 61.42 | 81.07 | 54.70 | 86.71 |
| ReAct [52] | 8.79 | **44.52** | **91.81** | 52.71 | 90.16 | 62.66 | 87.83 | 70.73 | 76.85 | 57.66 | 86.67 |
| SVD | 18.01 | 46.54 | 81.49 | **27.88** | **92.18** | 38.26 | **88.34** | **46.06** | **89.33** | 39.69 | **87.84** |
| PI (#100 iter) | 9.97 | 46.59 | 81.49 | 27.93 | 92.18 | 38.28 | 88.34 | 46.09 | 89.33 | 39.72 | **87.84** |
| PI (#50 iter) | 9.47 | 46.58 | 81.49 | 27.93 | 92.17 | **38.24** | 88.34 | 46.12 | 89.32 | 39.72 | 87.83 |
| PI (#20 iter) | 9.22 | 46.58 | 81.48 | 27.93 | 92.15 | 38.28 | 88.31 | 46.10 | 89.33 | 39.75 | 87.82 |
| PI (#10 iter) | 9.03 | 46.77 | 81.29 | 28.21 | 91.84 | 38.44 | 87.94 | 46.08 | 89.37 | 39.88 | 87.61 |
| PI (#5 iter) | 9.00 | 48.34 | 79.81 | 30.44 | 89.71 | 41.33 | 84.97 | 45.34 | 89.41 | 41.36 | 85.98 |

**The approximate solution by PI yields competitive performances.** Table 7 compares the time consumption and performance of SVD and PI, as well as two recent *state-of-the-art* OOD methods `ReAct` and `GradNorm`. The performance of PI starts to become competitive against that of SVD ($<0.1\%$) from **20** iterations on with **48.41%** time reduction. Compared with `ReAct`, the PI-based `RankFeat` only requires marginally $4.89\%$ more time consumption. `GradNorm` is not comparable against other baselines in terms of time cost because it does not support the batch mode.

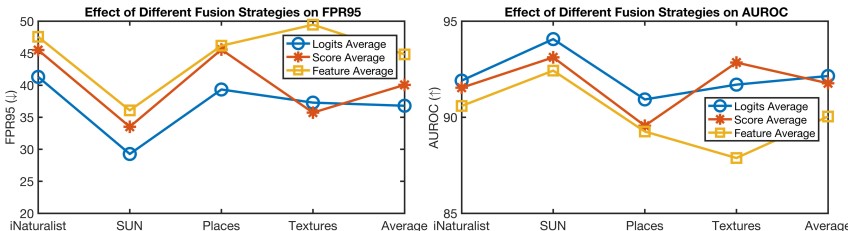

Figure 4: The impact of different fusion strategies on FPR95 and AUROC.

**Fusion at the logit space achieves the best performance.** Fig. 4 displays the performance of different fusion strategies in combining `RankFeat` at the Block 3 and Block 4 features. As can be

observed, averaging logits outperforms other fusion strategies in most datasets and metrics. This indicates that the fusing the logits can best coordinate the benefit of both features.

## 5 Related Work

**Distribution shifts.** Distribution shifts have been a long-standing problem in the machine learning research community [17, 45, 34, 62]. The problem of distributions shifts can be generally categorized as shifts in the input space and shifts in the label space. Shifts only in the input space are often deemed as *covariate shifts* [21, 44]. In this setting, the inputs are corrupted by perturbations or shifted by domains, but the label space stays the same [25, 53]. The aim is mainly to improve the robustness and generalization of a model [23]. For OOD detection, the labels are disjoint and the main concern is to determine whether a test sample should be predicted by the pre-trained model [39, 25].

Some related sub-fields also tackle the problem of distribution shifts in the label space, such as novel class discovery [16, 69], open-set recognition [48, 59], and novelty detection [1, 56]. These sub-fields target specific distribution shifts (*e.g.,* semantic novelty), while OOD encompasses all forms of shifts.

**OOD detection with discriminative models.** The early work on discriminative OOD detection dates back to the classification model with rejection option [4, 12]. The OOD detection methods can be generally divided into training-need methods and *post hoc* approaches. Compared with training-needed approaches, *post hoc* methods do not require any extra training processes and could be directly applied to any pre-trained models. For the wide body of research on OOD detection, please refer to [65] for the comprehensive survey. Here we only highlight the representative *post hoc* methods. Nguyen *et al.* [43] first observed the phenomenon that neural networks easily give over-confident predictions for OOD samples. The following researches attempted to improve the OOD uncertainty estimation by proposing ODIN score [39], OpenMax score [2], Mahalanobis distance [38], and Energy score [40]. Huang *et al.* [26] pointed out that the traditional CIFAR benchmark does not extrapolate to real-world settings and proposed a large-scale ImageNet benchmark. More recently, Sun *et al.* [52] and Huang *et al.* [27] proposed to tackle the challenge of OOD detection from the lens of activation abnormality and gradient norm, respectively. In contrast, based on the empirical observation of singular value distributions, we propose a simple yet effective *post hoc* solution by removing the rank-1 subspace from the high-level features.

**OOD detection with generative models.** Different from discriminative models, generative models detect the OOD samples by estimating the probability density function [32, 55, 47, 57, 8, 28, 3, 30]. A sample with a low likelihood is deemed as OOD data. Recently, a multitude of methods have utilized generative models for OOD detection [46, 51, 61, 64, 33, 49, 31]. However, as pointed out in [42], generative models could assign a high likelihood to OOD data. Furthermore, generative models can be prohibitively harder to train and optimize than their discriminative counterparts, and the performance is often inferior. This might limit their practical usage.

## 6 Conclusion

In this paper, we present `RankFeat`, a simple yet effective approach for *post hoc* OOD detection by removing the rank-1 matrix composed by the largest singular value from the high-level feature. We demonstrate its superior empirical results and the general applicability across architectures, network depths, and benchmarks. Extensive ablation studies and comprehensive theoretical analyses are conducted to reveal the important insights and to explain the working mechanism of our method.

## Acknowledgments and Disclosure of Funding

This research was supported by the EU H2020 projects AI4Media (No. 951911) and SPRING (No. 871245). We thank our colleague Zhun Zhong for the fruitful discussion and valuable suggestions.

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
