# RankFeat: Rank-1 Feature Removal for Out-of-distribution Detection –Supplementary Material–

## A Experimental Setup

**Implementation Details.** At the inference stage, all the images are resized to $480{\times}480$ for ResNetv2-101 [7] and SqueezeNet [12]. The source codes are implemented with `Pytorch 1.10.1`, and all experiments are run on a single NVIDIA Quadro RTX 6000 GPU.

**Evaluation Metrics.** Following [10, 19, 11], we measure the performance using two main metrics: (1) the false positive rate (FPR95) of OOD examples when the true positive rate of ID samples is at 95%; and (2) the area under the receiver operating characteristic curve (AUROC).

```
1  #Our RankFeat (SVD) is applied on each individual \\
2  #feature matrix within the mini-batch.
3  feat = model.features(inputs)
4  B, C, H, W = feat.size()
5  feat = feat.view(B, C, H * W)
6  u,s,vt = torch.linalg.svd(feat)
7  feat = feat - s[:,0:1].unsqueeze(2)*u[:,:,0:1].bmm(vt[:,0:1,:])
8  feat = feat.view(B,C,H,W)
9  logits = model.classifier(feat)
10 score = torch.logsumexp(logits, dim=1)
```

Figure 1: Pytorch-like codes of our `RankFeat` implementation.

**Pseudo Code of RankFeat.** Fig. 1 presents the Pytorch-like implementation of our `RankFeat`. We use `torch.linalg.svd` to conduct SVD on each individual feature matrix in the mini-batch.

## B More Evaluation Results

### B.1 Large-scale Species Dataset

The Species [9] dataset is a large-scale OOD validation benchmark consisting of $71,3449$ images, which is designed for ImageNet-1k [1] and ImageNet 21-k [13] as the ID sets. We select four sub-sets as the OOD benchmark, namely `Protozoa`, `Microorganisms`, `Plants`, and `Mollusks`. Table 1 present the evaluation results. Our `RankFeat` achieves the best performance, surpassing other methods by **15.91%** in the average FPR95 and by **3.31%** in the average AUROC.

### B.2 CIFAR100 with Different Architectures

We also evaluate our method on the CIFAR benchmark with various model architectures. The evaluation OOD datasets are the same with those of the ImageNet-1k benchmark. We take ResNet-56 [6] and RepVGG-A0 [2] pre-trained on ImageNet-1k as the backbones, and then fine-tune them on CIAR100 [14] for 100 epochs. The learning rate is initialized with $0.1$ and is decayed by 10 every

Table 1: The evaluation results on four sub-sets of Species [9] based on ResNetv2-101 [7]. All values are reported in percentages, and these *post hoc* methods are directly applied to the model pre-trained on ImageNet-1k [1]. The best three results are highlighted with red, blue, and cyan.

| Methods | Protozoa | | Microorganisms | | Plants | | Mollusks | | Average | |
|---|---|---|---|---|---|---|---|---|---|---|
| | FPR95 ($\downarrow$) | AUROC ($\uparrow$) | FPR95 ($\downarrow$) | AUROC ($\uparrow$) | FPR95 ($\downarrow$) | AUROC ($\uparrow$) | FPR95 ($\downarrow$) | AUROC ($\uparrow$) | FPR95 ($\downarrow$) | AUROC ($\uparrow$) |
| MSP [8] | 75.81 | 83.20 | 72.23 | 84.25 | 61.48 | 87.78 | 85.62 | 70.51 | 73.79 | 81.44 |
| ODIN [16] | 75.97 | 85.11 | 65.94 | 89.35 | 55.69 | 90.79 | 86.22 | 71.31 | 70.96 | 84.14 |
| Energy [17] | 79.49 | 84.34 | 60.87 | 90.30 | 54.67 | 90.95 | 88.47 | 70.53 | 70.88 | 84.03 |
| ReAct [19] | 81.74 | 84.26 | 58.82 | 85.88 | 36.90 | 93.78 | 90.58 | 76.33 | 67.02 | 85.06 |
| **RankFeat (Block 4)** | 66.98 | 70.19 | 39.06 | 86.67 | 46.31 | 79.98 | 80.14 | 59.92 | 58.12 | 74.19 |
| **RankFeat (Block 3)** | 58.99 | 88.81 | 49.72 | 90.04 | 47.01 | 91.85 | 80.37 | 79.61 | 59.02 | 87.58 |
| **RankFeat (Block 3 + 4)** | 52.78 | 88.65 | 37.21 | 92.82 | 38.07 | 92.88 | 76.38 | 78.13 | 51.11 | 88.37 |

Table 2: The evaluation results with different model architectures on CIFAR100 [14]. All values are reported in percentages, and these *post hoc* methods are directly applied to the model. The best two results are highlighted with red and blue.

| Model | Methods | iNaturalist | | SUN | | Places | | Textures | | Average | |
|---|---|---|---|---|---|---|---|---|---|---|---|
| | | FPR95 ($\downarrow$) | AUROC ($\uparrow$) | FPR95 ($\downarrow$) | AUROC ($\uparrow$) | FPR95 ($\downarrow$) | AUROC ($\uparrow$) | FPR95 ($\downarrow$) | AUROC ($\uparrow$) | FPR95 ($\downarrow$) | AUROC ($\uparrow$) |
| **RepVGG-A0 [2]** | MSP [8] | 61.55 | 85.03 | 91.05 | 69.19 | 65.45 | 82.10 | 86.68 | 65.56 | 76.18 | 75.47 |
| | ODIN [16] | 50.20 | 87.88 | 88.00 | 66.56 | 61.85 | 79.34 | 84.87 | 63.89 | 71.23 | 74.42 |
| | Energy [17] | 53.71 | 84.59 | 86.71 | 66.58 | 59.71 | 78.64 | 84.57 | 63.88 | 71.18 | 73.42 |
| | Mahalanobis [15] | 81.43 | 74.81 | 89.77 | 67.12 | 79.94 | 73.06 | 64.95 | 82.19 | 78.91 | 74.30 |
| | GradNorm [11] | 78.87 | 68.21 | 95.10 | 44.73 | 66.25 | 75.41 | 92.98 | 43.83 | 83.30 | 58.05 |
| | ReAct [19] | 48.09 | 93.00 | 73.87 | 78.12 | 61.63 | 78.43 | 75.23 | 81.36 | 64.71 | 82.73 |
| | **RankFeat** | 40.19 | 88.06 | 70.47 | 76.35 | 57.75 | 83.58 | 52.89 | 83.28 | 55.33 | 82.82 |
| **ResNet-56 [6]** | MSP [8] | 77.69 | 78.25 | 93.54 | 66.93 | 81.57 | 76.71 | 88.47 | 65.79 | 85.32 | 71.92 |
| | ODIN [16] | 66.92 | 79.25 | 95.05 | 50.45 | 77.45 | 72.88 | 90.51 | 53.47 | 82.48 | 64.01 |
| | Energy [17] | 65.24 | 79.13 | 95.05 | 49.33 | 77.10 | 72.32 | 90.39 | 52.68 | 81.95 | 63.37 |
| | Mahalanobis [15] | 89.47 | 69.32 | 91.38 | 54.76 | 82.32 | 77.53 | 68.83 | 79.64 | 83.00 | 70.31 |
| | GradNorm [11] | 96.72 | 42.09 | 94.19 | 47.97 | 94.61 | 48.09 | 89.14 | 50.18 | 93.67 | 47.08 |
| | ReAct [19] | 50.59 | 90.56 | 69.23 | 85.79 | 55.38 | 87.98 | 82.60 | 75.51 | 64.50 | 84.96 |
| | **RankFeat** | 34.62 | 88.21 | 61.82 | 80.50 | 53.79 | 89.71 | 30.89 | 91.31 | 45.28 | 87.43 |

30 epoch. Notice that this training process is to obtain a well-trained classifier but the ODO methods (including ours) are still *post hoc* and do not need any extra training.

Table 2 compares the performance against all the *post hoc* baselines. Our `RankFeat` establishes the *state-of-the-art* performances across architectures on most datasets and metrics, outperforming the second best method by **9.38 %** in the average FPR95 on RepVGG-A0 and by **19.22 %** in the average FPR95 on ResNet-56. Since the CIFAR images are small in resolution (*i.e.,* $32{\times}32$), the downsampling times and the number of feature blocks of the original models are reduced. Hence we only apply `RankFeat` to the final feature before the last GAP layer.

## B.3 One-class CIFAR10

To further demonstrate the applicability of our method, we follow [3, 4, 20] and conduct experiments on one-class CIFAR10. The setup is as follows: we choose one of the classes as the ID set while keeping other classes as OOD sets. Table 3 reports the average AUROC on CIFAR10. Our `RankFeat` outperforms other baselines on most sub-set as well as on the average result.

Table 3: The average AUROC (%) on one-class CIFAR10 based on ResNet-56.

| Methods | Plane | Car | Bird | Cat | Deer | Dog | Frog | Horse | Ship | Truck | Mean |
|---|---|---|---|---|---|---|---|---|---|---|---|
| MSP | 59.75 | 52.48 | 62.96 | 48.73 | 59.15 | 52.39 | 67.33 | 59.34 | 54.55 | 51.97 | 56.87 |
| Energy | 83.12 | 91.56 | 68.99 | 56.02 | 75.03 | 77.33 | 69.50 | 88.41 | 82.88 | 84.74 | 77.76 |
| ReAct | 82.24 | 96.69 | 78.32 | 76.84 | 76.11 | 86.80 | 86.15 | 90.95 | 89.91 | 94.17 | 85.82 |
| **RankFeat** | 79.26 | 98.54 | 82.04 | 80.28 | 82.89 | 90.28 | 89.06 | 95.30 | 94.11 | 94.02 | 88.58 |

## C   Baseline Methods

For the convenience of audiences, we briefly recap the previous *post hoc* methods for OOD detection. Some implementation details of the methods are also discussed.

**MSP [8].** One of the earliest work considered directly using the Maximum Softmax Probability (MSP) as the scoring function for OOD detection. Let $f(\cdot)$ and $\mathbf{x}$ denote the model and input, respectively. The MSP score can be computed as:

$$\texttt{MSP}(\mathbf{x}) = \max\Big(\text{Softmax}(f(\mathbf{x}))\Big) \tag{1}$$

Despite the simplicity of this approach, the MSP score often fails as neural networks could assign arbitrarily high confidences to the OOD data [18].

**ODIN [16].** Based on MSP [8], ODIN [16] further integrated temperature scaling and input perturbation to better separate the ID and OOD data. The ODIN score is calculated as:

$$\texttt{ODIN}(\mathbf{x}) = \max\Big(\text{Softmax}(\frac{f(\bar{\mathbf{x}})}{T})\Big) \tag{2}$$

where $T$ is the hyper-parameter temperature, and $\bar{\mathbf{x}}$ denote the perturbed input. Following the setting in [11], we set $T{=}1000$. According to [11], the input perturbation does not bring any performance improvement on the ImageNet-1k benchmark. Hence, we do not perturb the input either.

**Energy score [17].** Liu *et al.* [17] argued that an energy score is superior than the MSP because it is theoretically aligned with the input probability density, *i.e.,* the sample with a higher energy correspond to data with a lower likelihood of occurrence. Formally, the energy score maps the logit output to a scalar function as:

$$\texttt{Energy}(\mathbf{x}) = \log \sum_{i=1}^{C} \exp(f_i(\mathbf{x})) \tag{3}$$

where $C$ denotes the number of classes.

**Mahalanobis distance [15].** Lee *et al.* [15] proposed to model the Softmax outputs as the mixture of multivariate Gaussian distributions and use the Mahalanobis distance as the scoring function for OOD uncertainty estimation. The score is computed as:

$$\texttt{Mahalanobis}(\mathbf{x}) = \max_i \Big( -(f(\mathbf{x}) - \mu_i)^T \Sigma (f(\mathbf{x}) - \mu_i) \Big) \tag{4}$$

where $\mu_i$ denotes the feature vector mean, and $\Sigma$ represents the covariance matrix across classes. Following [11], we use 500 samples randomly selected from ID datasets and an auxiliary tuning dataset to train the logistic regression and tune the perturbation strength $\epsilon$. For the tuning dataset, we use FGSM [5] with a perturbation size of 0.05 to generate adversarial examples. The selected $\epsilon$ is set as 0.001 for ImageNet-1k.

**GradNorm [11].** Huang *et al.* [11] proposed to estimate the OOD uncertainty by utilizing information extracted from the gradient space. They compute the KL divergence between the Softmax output and a uniform distribution, and back-propagate the gradient to the last layer. Then the vector norm of the gradient is used as the scoring function. Let $\mathbf{w}$ and $\mathbf{u}$ denote the weights of last layer and the uniform distribution. The score is calculated as:

$$\texttt{GradNorm}(\mathbf{x}) = ||\frac{\partial D_{KL}(\mathbf{u}||\text{Softmax}(f(\mathbf{x})))}{\partial \mathbf{w}}||_1 \tag{5}$$

where $||\cdot||_1$ denotes the $L_1$ norm, and $D_{KL}(\cdot)$ represents the KL divergence measure.

**ReAct [19].** In [19], the authors observed that the activations of the penultimate layer are quite different for ID and OOD data. The OOD data is biased towards triggering very high activations, while the ID data has the well-behaved mean and deviation. In light of this finding, they propose to clip the activations as:

$$f_{l-1}(\mathbf{x}) = \min(f_{l-1}(\mathbf{x}), \tau) \tag{6}$$

where $f_{l-1}(\cdot)$ denotes the activations for the penultimate layer, and $\tau$ is the upper limit computed as the 90-th percentile of activations of the ID data. Finally, the Energy score [17] is computed for estimating the OOD uncertainty.

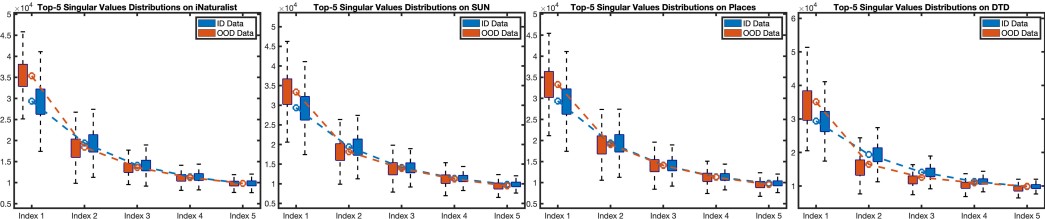

Figure 2: The top-5 singular value distribution of the ID dataset and OOD datasets. The first singular values $s_1$ of OOD data are consistently much larger than those of ID data on each OOD dataset.

# D    Visualization about RankFeat

## D.1    Singular Value Distribution

Fig. 2 compares the top-5 singular value distribution of ID and OOD feature matrices on all the datasets. Our novel observation consistently holds for every OOD dataset: the dominant singular value $s_1$ of OOD feature always tends to be significantly larger than that of ID feature.

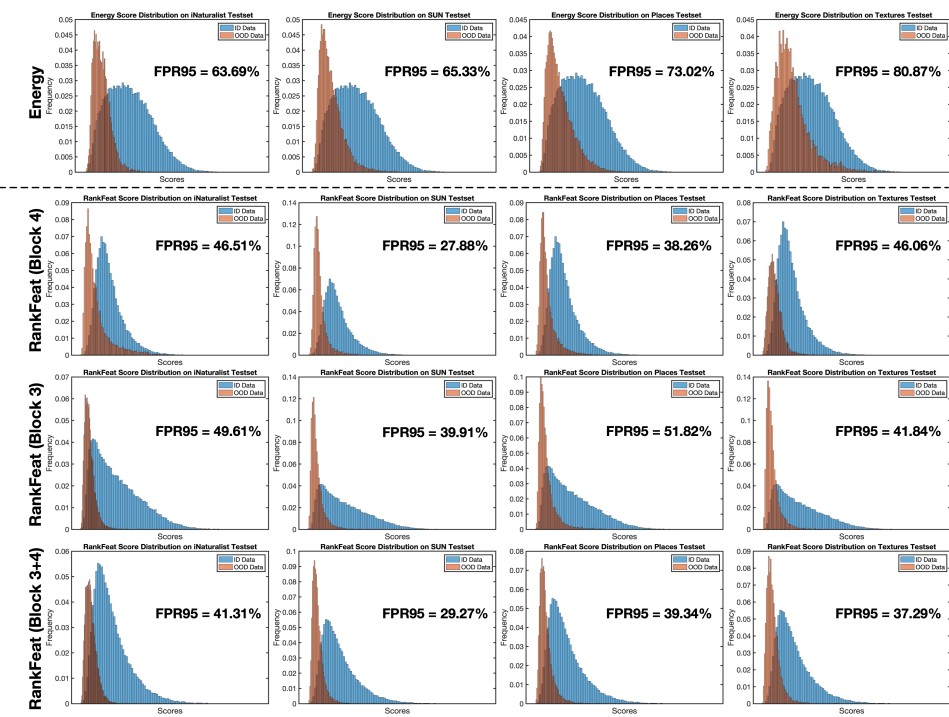

Figure 3: The score distributions of `Energy` [17] (top row) and our proposed `RankFeat` (rest rows) on four OOD datasets. Our `RankFeat` applies to different high-level features at the later depths of the network, and their score functions can be further fused.

## D.2    Score Distribution

Fig. 3 displays the score distributions of `RankFeat` at Block 3 and Block 4, as well as the fused results. Our `RankFeat` works for both high-level features. For the score fusion, when Block 3 and Block 4 features are of similar scores ($diff. < 5\%$), the feature combination could have further improvements.

### D.3 Output Distribution

Fig. 4(a) presents the output distribution (*i.e.,* the logits after `Softmax` layer) on `ImageNet` and `iNaturalist`. After our `RankFeat`, the OOD data have a larger reduction in the probability output; most of OOD predictions are of very small probabilities ($<0.1$).

### D.4 Logit Distribution

Fig. 4(b) displays the logits distribution of our `RankFeat`. The OOD logits after `RankFeat` have much less variations and therefore are closer to the uniform distribution.

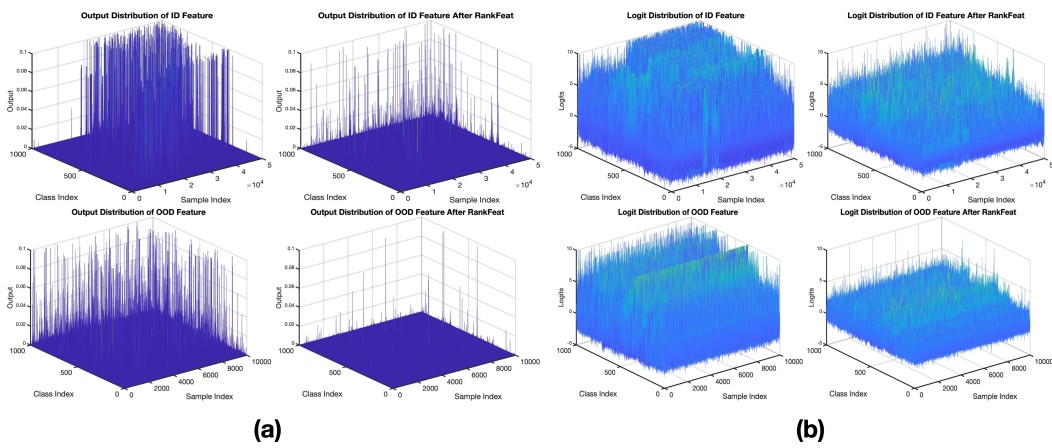

Figure 4: **(a)** Output distributions of `RankFeat`. **(b)** Logit distributions of `RankFeat`.

## E  Why are the singular value distributions of ID and OOD features different?

In the paper, we give some theoretical analysis to explain the working mechanism of our `RankFeat`. It would be also interesting to investigate why the singular value distributions of the ID and OOD features are different. Here we give an intuitive conjecture. Since the network is well trained on the ID training set, when encountered with ID data, the feature matrix is likely to be more informative. Accordingly, more singular vectors would be active and the matrix energies spread over the corresponding singular values, leading to a more flat spectrum. On the contrary, for the unseen OOD data, the feature is prone to have a more compact representation, and less singular vectors might be active. In this case, the dominant singular value of OOD feature would be larger and would take more energies of the matrix. The informativeness can also be understood by considering applying PCA on the feature matrix. Suppose that we are using PCA to reduce the dimension of ID and OOD feature to $1$. The amount of retained information can be measured by explained variance (%). The metric is defined as $\sum_{i=0}^{k} \mathbf{s}_i^2 / \sum_{j=0}^{n} \mathbf{s}_j^2$ where $k$ denotes the projected dimension and $n$ denotes the total dimension. It measures the portion of variance that the projected data could account for. We compute the average explained variance of all datasets and present the result in Table 4.

Table 4: The average explained variance ratio (%) of the ID and OOD datasets.

| Dataset | ImageNet-1k | iNaturalist | SUN | Places | Textures |
|---|---|---|---|---|---|
| Explained Variance (%) | **28.57** | 38.74 | 35.79 | 35.17 | 42.21 |

As can be observed, the OOD datasets have a larger explained variance ratio than the ID dataset. *That being said, to retain the same amount of information, we need fewer dimensions for the projection of OOD features. This indicates that the information of OOD feature is easier to be captured and the OOD feature matrix is thus less informative.*

*As for how the training leads to the difference, we doubt that the well-trained network weights might cause and amplify the gap in the dominant singular value of the ID and OOD feature.* To verify this guess, we compute the singular values distributions of the Google BiT-S ResNetv2-100 model [7, 13] with different training steps, as well as a randomly initialized network as the baseline.

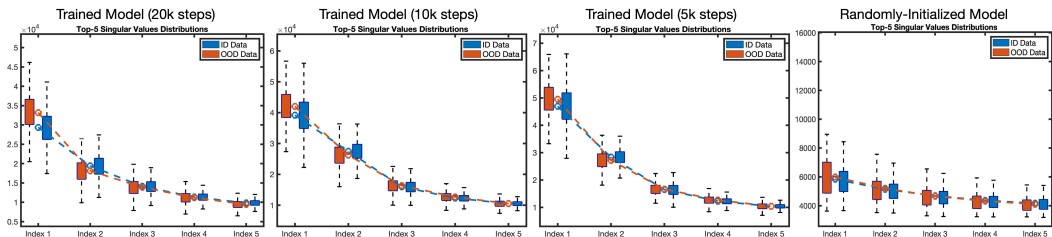

Figure 5: The top-5 largest singular value distributions of the pre-trained network with different training steps. For the untrained network initialized with random weights, the singular values distributions of ID and OOD feature exhibit very similar behaviors. As the training step increases, the difference between the largest singular value is gradually amplified.

Fig. 5 depicts the top-5 largest singular value distributions of the network with different training steps. Unlike the trained networks, the untrained network with random weights has quite a similar singular value distribution for the ID and OOD data. The singular values of both ID and OOD features are of similar magnitudes with the untrained network. However, when the number of training steps is increased, the gap of dominant singular value between ID and OOD feature is magnified accordingly. This phenomenon supports our conjecture that the well-trained network weights cause and amplify the difference of the largest singular value. Interestingly, our finding is coherent with [21]. In [21], the authors demonstrate that the classification accuracy of a model is highly correlated with its ability of OOD detection and open-set recognition. Training a stronger model could naturally improve the OOD detection performance. We empirically show that the gap of the dominant singular value is gradually amplifying as the training goes on, which serves as supporting evidence for [21].

## F  Theorem and Proof of Manchenko-Pastur Law

In the paper, we use the MP distribution of random matrices to show that removing the rank-1 matrix makes the statistics of OOD features closer to random matrices. For self-containment and readers' convenience, here we give a brief proof of Manchenko-Pastur Law.

**Theorem 1.** *Let $\mathbf{X}$ be a random matrix of shape $t \times n$ whose entries are random variables with $E(\mathbf{X}_{ij} = 0)$ and $E(\mathbf{X}_{ij}^2 = 1)$. Then the eigenvalues of the sample covariance $\mathbf{Y} = \frac{1}{n}\mathbf{X}\mathbf{X}^T$ converges to the probability density function: $\rho(\lambda) = \frac{t}{n}\frac{\sqrt{(\lambda_+ - \lambda)(\lambda - \lambda_-)}}{2\pi\lambda\sigma^2}$ for $\lambda \in [\lambda_-, \lambda_+]$ where $\lambda_- = \sigma^2(1 - \sqrt{\frac{n}{t}})^2$ and $\lambda_+ = \sigma^2(1 + \sqrt{\frac{n}{t}})^2$.*

*Proof.* Similar with the deduction of our bound analysis, the sample covariance $\mathbf{Y}$ can be written as the sum of rank-1 matrices:

$$\mathbf{Y} = \sum_{s=0}^{t} = \mathbf{Y}_n^s, \; \mathbf{Y}_n^s = \mathbf{U}_n^s \mathbf{D}_n^s (\mathbf{U}_n^s)^* \tag{7}$$

where $\mathbf{U}_n^s$ is a unitary matrix, and $\mathbf{D}_n^s$ is a diagonal matrix with the only eigenvalue $\beta = n/t$ for large $n$ (rank-1 matrix). Then we can compute the Stieltjes transform of each $\mathbf{Y}_n^s$ as:

$$s_n(z) = \frac{1}{n}\mathrm{tr}(\mathbf{Y}_n^s - z\mathbf{I})^{-1} \tag{8}$$

Relying on Neumann series, the above equation can be re-written as:

$$s_n(z) = -\frac{1}{n}\sum_{k=0}^{\infty}\frac{\mathrm{tr}(\mathbf{Y}_n^s)^t}{z^{k+1}} = -\frac{1}{n}\left(\frac{n}{z} + \sum_{k=1}^{\infty}\frac{\beta^k}{z^{k+1}}\right) = -\frac{1}{n}\left(\frac{n-1}{z} + \frac{1}{z-\beta}\right) \tag{9}$$

Let $z := z_n(s)$ and we can find the function inverse of the transform:

$$nsz_n(s)^2 - n(s\beta - 1)z_n(s) - (n-1)\beta = 0 \tag{10}$$

The close-formed solution is calculated as:

$$z_n(s) = \frac{n(s\beta - 1) \pm \sqrt{n^2(s\beta - 1)^2 + 4n(n-1)s\beta}}{2ns}$$

$$\approx \frac{1}{2ns}\left(n(s\beta - 1) \pm \left| n(s\beta + 1) - \frac{2s\cancel{\beta}}{\cancel{\beta}+1} \right| \right) \tag{11}$$

For large $n$, the term $\frac{2s\beta}{\beta+1}$ is sufficiently small and we can omit it. The solution is defined as:

$$z_n(s) = -\frac{1}{s} + \frac{\beta}{n(1 + s\beta)} \tag{12}$$

The R transform of each $\mathbf{Y}_n^s$ is given by:

$$R_{\mathbf{Y}_n^s}(s) = z_n(-s) - \frac{1}{s} = \frac{\beta}{n(1 - s\beta)} \tag{13}$$

Accordingly, the R transform for $\mathbf{Y}_n$ is given by:

$$R_{\mathbf{Y}}(s) = tR_{\mathbf{Y}_n^s}(s) = \frac{\beta t}{n(1 - s\beta)} = \frac{1}{1 - s\beta} \tag{14}$$

Thus, the inverse Stieltjes transform of $\mathbf{Y}$ is

$$z(s) = -\frac{1}{s} + \frac{1}{1 + s\beta} \tag{15}$$

Then the Stieltjes transform of $\mathbf{Y}$ is computed by inverting the above equation as:

$$s(z) = \frac{-(z + \beta + 1) + \sqrt{(z + \beta + 1)^2 - 4\beta z}}{2z\beta} \tag{16}$$

Since $\beta = {}^b/t$, finding the limiting distribution of the above equation directly gives the Manchenko-Pastur distribution:

$$\rho(\lambda) = \frac{t}{n} \frac{\sqrt{(\lambda_+ - \lambda)(\lambda - \lambda_-)}}{2\pi\lambda\sigma^2} \; for \; \lambda \in [\lambda_-, \lambda_+],$$

$$\lambda_- = \sigma^2(1 - \sqrt{\frac{n}{t}})^2, \lambda_+ = \sigma^2(1 + \sqrt{\frac{n}{t}})^2 \tag{17}$$

The theorem is thus proved. $\qquad\qquad\square$