# OpenReview forum: "RankFeat: Rank-1 Feature Removal for Out-of-distribution Detection"
_NeurIPS.cc/2022/Conference — NeurIPS 2022 Accept_

### Official Review · Reviewer_SMeo · 2022-06-15

**Rating:** 5
**Confidence:** 4
**Soundness:** 1 poor
**Presentation:** 3 good
**Contribution:** 1 poor

**Summary:**

This paper aims to address OOD detection issue by proposing a novel method called RankFeat.

By observing the OOD feature matrix tends to have a significantly larger dominant singular value, the author believes that by removing the rank-1 feature, the over-confidence of OOD samples is mitigated, and consequently the ID and OOD data can be better distinguished.

Based on the above idea, the propose RankFeat, which aims to removing the rank-1 matrix from the high-level feature.

Theory has shown that it seems RankFeat achieves smaller energy score than ReAct.

Additionally, experiments have shown this method achieve a large improvement.

Lastly, ablation studies are conducted.

**Questions:**

The basic idea of this paper does give researchers a novel perspective to understand OOD detection. I also like this idea. But I have many issues raised.

Issues:
(Major)1. In the real world applications, the OOD datasets can be changeable and diverse. Could you guarantee the observation in Figure 1(a) still holds? I would very, very much doubt this observation. I think you should conduct much more experiments to support the basic view. In fact, if you can give theory (allowed assumptions) or abundant experiments to support the basic view, I think this paper can be accepted by NeurIPS; otherwise, I may not accept it.

(Major) 2. Experiments should be conducted to show why energy-based score is optimal for your method in Eq. 9.

(Minor) 3. In line 222, I find you remove features from 1 to n. How about only removing features n?

(Major) 4. The theory is over-claimed. Please revise it according to weakness 4 and 5.

I think this paper can be accepted if the authors can address all my issues.





**Limitations:**

In the real world applications, the OOD datasets can be changeable and diverse. Could you guarantee the observation in Figure 1(a) still holds? I would very, very much doubt this observation. I think you should conduct much more experiments to support the basic view. In fact, if you can give theory (allowed assumptions) or abundant experiments to support the basic view, I think this paper can be accepted by NeurIPS; otherwise, I may not accept it.

**Strengths And Weaknesses:**

Strengths:
1. This method is easy and the idea is clear.
2. The writing is good and organisation is also good.
3. Experiments are rich.

The basic idea of this paper does give researchers a novel perspective to understand OOD detection. I also like this idea.

Weakness:
1. I still can not understand figure 1(b). Could you provide more detailed introduction?
2. I am interesting in more datasets to conduct experiments in figure 1(a).
3. According to Eq. 9, your method is based on an assumption that the energy-based score is effective. Do you have checked other score function to implement your idea? or do you have argued that energy-based score is the optimal for your method?
4. The theory is not right The basic reason is that you cannot guarantee the bound in Eq 17 is the tightest bound, so you cannot use it to compare with your method's bound Eq 14. For example, a<1, b<2 cannot conclude that a<b. Here you have the same mistake.
5. The theory is weak. Eq 14 has shown that  RankFeat achieves small energy score . However, you omit a basic issue that why  a small energy score is good for OOD detection. You should use theory to explain it; otherwise, I think your paper has some unmentioned assumptions.

---

> ### Author Response · Authors · 2022-08-01
> **Response to Reviewer SMeo (Part 1/1); Thanks for your encouraging feedback and constructive suggestions!**
>
> We thank $\color{brown}{Reviewer\ SMeo\ (R4)}$ for the encouraging feedback and the constructive suggestions! The response to the concerns are as follows:
>
> ___
>
> **1. Detailed Explanation of Fig. 1(b)**
>
> Thanks for the careful review! For Fig. 1(b), the x-axis is the sample index and the y-axis denotes the index of the predicted class in the range from $1$ to $1,000$ (ImageNet-1k has $1,000$ categories). The orange points represent the original class prediction, while the blue points are the class prediction when our $\texttt{RankFeat}$ is applied. For OOD data, the orange and blue points hardly overlap, which indicates that our RankFeat largely perturbs the decision. By contrast, many decisions of ID data stay consistent, which is reflected by the heavily-overlapped straight lines. *The phenomenon could imply that the OOD decisions are more dependent on the rank-1 feature matrix*.
>
> **2. More Experiments of Fig. 1(a)**
>
> Thanks for the constructive suggestion. ***The experiment in Fig. 1 is based on SUN, but the observation holds for every OOD dataset***. In Sec. D.1 of the revised supplementary material, we have added the top-5 singular value distribution of all the OOD datasets. The observation is consistent on every OOD dataset: the first singular value of the OOD feature always tends to be much larger than that of ID feature.
>
> **3. Why We Use Energy Score**
>
> Thanks for the insightful advice! We choose Energy score because it is both empirically and theoretically sound. Empirically, it achieves better performance than other basic score functions like MSP, ODIN, and Mahalanobis distance (see Table 2 of the paper). Theoretically, it aligns with the probability density of the input: the energies of samples are associated with the likelihood of occurrence of data. We have added the reason to line120 (Eq. (9)) of the revised paper.
>
> This can be explained from our upper bound analysis. The basis of the theory is the tight bound $\max(\mathbf{y}')   < \log \sum \exp(\mathbf{y}') < \max(\mathbf{y}') + \log(Q)$, which is brought by the Log-Sum-Exp trick. *This is an inherent property of Energy score. If we use other score functions, the inequality and subsequent bound analysis might not hold*.
>
> **4. Bound Analysis is to Give Insight**
>
> Thanks for the thoughtful comment! ***We would like to clarify that our upper bound analysis strives to give insights into the working mechanism of RankFeat instead of giving a strict guarantee of the score***. Considering the diversity of model architectures and data distributions, it might be unlikely to give a strong and generic theoretical guarantee. *Our bound analysis strives to improve the understanding of OOD methods from new perspectives*. We have shown several use cases of our bound analysis:
> - The impact of removing the rank-1 matrix can be analyzed concretely by studying the upper bound (see line129).
> - Relying on our bound analysis, we can build a theoretical connection between our RankFeat and ReAct (see line157).
> - The bound analysis can be used to explain the shrinkage and skew of score distributions in Fig. 2 (see line134).
>
> **5. Performance of Removing $\mathbf{s_{i}}\mathbf{u_{i}}\mathbf{v_{i}}^{T}$ for $i>1$**
>
> Thanks for the insightful question. We did one experiment of removing $\mathbf{s_{i}}\mathbf{u_{i}}\mathbf{v_{i}}^{T}$ for $i=2$ and $i=3$ , and present the results in the format of FPR95 ($\downarrow$) / AUROC ($\uparrow$) below:
>
> |Method|iNaturalist|SUN|Places|Textures|
> |-|:-:|:-:|:-:|:-:|
> |Energy|64.91/88.48|65.33/85.32|73.02/81.37|80.87/75.79|71.03/82.74|
> |Removing $\mathbf{s_{2}}\mathbf{u_{2}}\mathbf{v_{2}}^{T}$|62.12/89.28|65.84/85.23|73.33/80.88|80.41/76.45|70.42/82.96|
> |Removing $\mathbf{s_{3}}\mathbf{u_{3}}\mathbf{v_{3}}^{T}$|63.51/88.99|65.05/85.47|72.64/81.37|79.47/76.92|70.17/83.19|
>
> As can be seen above, the performance only gets marginal improvements over Energy score. *It meets our expectation as the other singular values of ID and OOD features are not that different*.
>
> ___
>
> Thanks again and it is more than welcome to post any further comments.

---

> > ### Comment · Reviewer_SMeo · 2022-08-03
> > **To Response**
> >
> > Dear Authors,
> >
> > I have read your response.
> >
> > To response 4, I am interesting in how you revise your theoretical part. I except your revision.
> >
> > Could you give a detailed explainations to response why the first singular value of the OOD feature always tends to be much larger than that of ID feature?
> >
> > If you answer these two issues, I will increase your score to 7.
> >
> > Best Regards,

---

> > > ### Author Response · Authors · 2022-08-03
> > > **Thanks for your instant reply and constructive suggestions.**
> > >
> > > We thank $\color{brown}{Reviewer\ SMeo}\ (R4)$ for the instant reply and constructive suggestions. Below we address the concerns in detail.
> > >
> > > ___
> > >
> > > **1. Revision of Theoretical Part**
> > >
> > > Regarding the revision of the theoretical part of Sec. 3, we try to make it more clear that the aim of our bound is to give insights. Specifically, we did:
> > >
> > > - We avoid deterministic tone when analyzing the impact of our upper bound. Instead, we use milder phrases like *'is likely to'* and *'would'*. The exemplary usages can be found at line109, line132, and line142 of the revised paper.
> > > - After we introduce our bound analysis, we immediately emphasize that ***“Notice that our bound analysis strives to improve the understanding of OOD methods from new perspectives instead of giving a strict guarantee of the score”*** at line134 of the revised paper. Then we present one use case of using bound analysis to explain the shrinkage and skewness of score distributions.
> > >
> > > We hope that these revisions can give the impression that our bound could give insights to a better understanding of our method.
> > >
> > >
> > > **2. Why OOD Feature Has a Larger $\mathbf{s_{1}}$**
> > >
> > > Thanks for the insightful advice! This behavior can be explained intuitively.
> > >
> > > ***Since the model is well-trained on the training set of ID data, the feature matrix is likely to be more informative when the model is fed with a similar validation set of ID data. On the contrary, the feature matrix tends to be less informative for the unseen OOD data***.
> > >
> > > The informativeness is directly reflected by the Eigen spectrum of Fig. 2 of the supplementary material: *an isolated larger dominant eigenvalue usually implies that the matrix is less informative*. This can be better understood by considering applying PCA to the feature matrix. Suppose that we are using PCA to reduce the dimension of ID and OOD features to 1. The amount of retained information can be measured by explained variance (%). The metric is defined as $\sum_{i=0}^{k}\mathbf{s_{i}}^2/\sum_{j=0}^{n}\mathbf{s_{j}}^2$ where  $k$ denotes the projected dimension and  $n$ denotes the total dimension. It measures the portion of variance that the projected data could account for. We compute the average explained variance of all datasets and present the result below:
> > >
> > > |Dataset |	ImageNet-1k |	iNaturalist | SUN | Places | Textures|
> > > |:--|:-:|:-:|:-:|:-:|:-:|
> > > |Explained Variance (%) |	28.57 |	38.74 |	35.79 |	35.17 |	42.21|
> > >
> > > As can be seen above, all the OOD datasets have a larger explained variance ratio than the ID dataset. *That being said, to retain the same amount of information, we need fewer dimensions for the projection of OOD features. This indicates that the information of the OOD feature is easier to be captured and the OOD feature matrix is thus less informative*.
> > >
> > > For a detailed explanation and analysis, please refer to Sec. E of the revised supplementary material.
> > > ___
> > >
> > > Thanks again and it is more than welcome to post any comments!

---

> > > > ### Comment · Reviewer_SMeo · 2022-08-05
> > > > **To Response**
> > > >
> > > > Why does the feature matrix tend to be less informative for the unseen OOD data?
> > > >
> > > > Consider the near OOD detection (A Simple Fix to Mahalanobis Distance for Improving Near-OOD Detection), the ID and OOD data may have overlap or be similar.
> > > >
> > > > In the near OOD detection case, could you guarantee this?
> > > >
> > > > I hope more experiments on the near OOD can be conducted to prove this.

---

> > > > > ### Author Response · Authors · 2022-08-05
> > > > > **Thanks for the follow-up; We have near-OOD experiments of CIFAR10 versus CIFAR100.**
> > > > >
> > > > > We thank $\color{brown}{Reviewer\ SMeo}$ for the follow-up and the insightful comments. Below we address the concerns in detail.
> > > > > ___
> > > > >
> > > > > **1. Why does the feature matrix tend to be less informative for the unseen OOD data?**
> > > > >
> > > > >
> > > > > For the unseen OOD data, the distribution is different from that of ID data. The feature matrix of OOD data might embed less information because the model weights never encounter the OOD distribution and are unaware of the OOD data. *This is reflected in the larger PCA explained variance ratio of OOD feature*.
> > > > >
> > > > > We totally understand the reviewer's concern that this is an intuitive explanation. However, we would like to note that most *post hoc* methods also have similar intuitions:
> > > > >
> > > > > - $\texttt{MSP}$ assumes that the OOD data correspond to a less determined prediction score.
> > > > > - $\texttt{Energy}$  assumes that the energy of the OOD logit is different from that of ID.
> > > > > - $\texttt{ReAct}$  assumes that the OOD feature has a larger abnormal value.
> > > > > - $\texttt{GradNorm}$  assumes that the gradient of OOD data has a larger vector norm.
> > > > >
> > > > > All of those methods intuitively assume the behavior of OOD and ID data is different in certain aspects. Our $\texttt{RankFeat}$ has a similar intuition with $\texttt{ReAct}$.
> > > > >
> > > > >
> > > > > **1. Near-OOD experiment of CIFAR10 versus CIFAR100**
> > > > >
> > > > > Thanks for the advice on the near-OOD evaluation. We agree that the near-OOD case where the ID and OOD are similar is indeed challenging to all *post hoc* methods because no prior knowledge is given.
> > > > >
> > > > > We have added a standard near-OOD experiment of CIFAR10 versus CIFAR100 in Sec. B.4 of the revised supplementary material, which is the same near-OOD benchmark as in [1]. Here we attach the result for the reviewer's convenience.
> > > > >
> > > > > |Method  | ID: CIFAR10 & OOD: CIFAR100 |	ID: CIFAR100 & OOD: CIFAR10|
> > > > > |:-:|:-:|:-:|
> > > > > |MSP |82.07/**78.27** |	86.86/70.54|
> > > > > |ODIN |	76.96/77.79 |	86.15/72.13|
> > > > > |Energy|	76.45/77.82 |	86.20/71.99|
> > > > > |ReAct|	79.48/71.80 |	88.41/69.86|
> > > > > |GradNorm|	84.45/52.17 |	92.78/52.19|
> > > > > |RankFeat|	**75.10**/78.02 |	**82.63**/**74.35**|
> > > > >
> > > > > The above Table reports the average FPR95  ($\downarrow$) / AUROC ($\uparrow$). As can be observed, our $\texttt{RankFeat}$ still has an advantage over other baselines in most metrics. ***This demonstrates that in the near-OOD detection case, the novel observation holds and our method still works***.
> > > > >
> > > > > >[1] A Simple Fix to Mahalanobis Distance for Improving Near-OOD Detection. ICML Workshop 2021
> > > > > ___
> > > > >
> > > > > Thanks again and it is more than welcome to post any further comments!

---

> > > > > > ### Comment · Reviewer_SMeo · 2022-08-05
> > > > > > **To Response**
> > > > > >
> > > > > > Thank you for your reply.
> > > > > >
> > > > > >  I still cannot understand why the first singular value of the OOD feature always tends to be much larger than that of ID feature.
> > > > > >
> > > > > > Because as I know the OOD is diverse and we cannot guarantee whether the OOD data 's first singular value is larger or smaller than that of ID.
> > > > > >
> > > > > > Could you conduct experiments for this case that
> > > > > >
> > > > > > 1. double circles: the insider circle is ID and the outside circle is OOD. At this case,  the first singular value of the OOD feature always tends to be much larger than that of ID feature?
> > > > > >
> > > > > > 2. double circles: the outsider circle is ID and the inside circle is OOD. At this case,  the first singular value of the OOD feature always tends to be much larger than that of ID feature?
> > > > > >
> > > > > > If 1 is correct, then 2 is not correct; if 2 is correct, then 1 is not correct. So your claim that the first singular value of the OOD feature always tends to be much larger than that of ID feature may be not correct.
> > > > > >
> > > > > > Could you explain above cases?
> > > > > >
> > > > > > If you answer above cases, I will add your scores to 7; otherwise, I will decrease your score.
> > > > > >
> > > > > > You may wander that why I am so care about this question? Because the question is the core discovery of your paper.

---

> > > > > > > ### Author Response · Authors · 2022-08-05
> > > > > > > **Thanks for the detailed explanation; Now we fully understand your question!**
> > > > > > >
> > > > > > > Thanks for your detailed explanation! Now we fully understand your question.
> > > > > > >
> > > > > > > ___
> > > > > > >
> > > > > > > In your described case 1 and case 2, the conclusion seems to be always contradictory. If case 1 holds then case 2 does not hold and vice versa. **However, one important thing that is missed here is that the models in case 1 and case 2 are totally different: the model is always pre-trained on the ID data.** So in case 1 the model is pre-trained on the insider circle, and the model in case 2 is pre-trained on the outsider circle. Since the two models are trained on different circles, case 1 and case 2 are not contradictory, and they can both hold.
> > > > > > > Actually, as we explained and empirically showed in Sec. E of the revised supplementary material (Figure 5), *it is the well-trained model weights that caused and amplified the first singular value difference of the ID and OOD features.*
> > > > > > >
> > > > > > > ___
> > > > > > >
> > > > > > > Hope that we understand your question correctly and that the above explanation is sufficient to eliminate your doubt. If not, please let us know and we will add more figures and data to explain this better. 😀

---

> > > > > > > > ### Comment · Reviewer_SMeo · 2022-08-05
> > > > > > > > **To Response**
> > > > > > > >
> > > > > > > > After a pre-trained model is given for ID, can we construct an OOD data to ensure that
> > > > > > > >
> > > > > > > > the first singular value of the OOD feature always tends to be much smaller than that of ID feature ?
> > > > > > > >
> > > > > > > > Or we can only construct OOD data whose  first singular value  is larger than ID?
> > > > > > > >
> > > > > > > > If the second case is correct, could you prove it?

---

> > > > > > > > > ### Author Response · Authors · 2022-08-05
> > > > > > > > > **Thank you for the follow-up and inspiration!**
> > > > > > > > >
> > > > > > > > > Thanks for your inspiration for future work!
> > > > > > > > >
> > > > > > > > > ___
> > > > > > > > >
> > > > > > > > > Given a pre-trained model and ID data, if we have statistics of the singular values distributions of the ID feature, it might be possible to cherry-pick OOD samples and collect an OOD dataset that might violate our observation (*the assumption of this is the prior knowledge of statistics of ID feature matrices and the cherry-picking process of OOD samples*). However, in practice for the existing OOD datasets that we have tested, we can empirically observe that the first singular value of OOD data always tends to be larger.
> > > > > > > > >
> > > > > > > > > Unfortunately, we cannot prove that it is always true in theory because the deep networks are highly non-linear and the training data can be very large-scale. The impact of a training (optimization) process on the model weights and on the feature statistics cannot be tractably analyzed in equations. We just searched the relevant literature again and we could not find any theoretical reference that did similar proofs. ***So we believe that finding a strong theoretical guarantee of (*post hoc*) OOD methods is still an open problem. It is a good and important research direction for our future work and even for the community.***
> > > > > > > > >
> > > > > > > > > ___
> > > > > > > > >
> > > > > > > > > Thanks again and it is more than welcome to post any further comments!

---

> > > > > > > > > > ### Comment · Reviewer_SMeo · 2022-08-05
> > > > > > > > > > **To Response**
> > > > > > > > > >
> > > > > > > > > > Thank your for your efforts.
> > > > > > > > > >
> > > > > > > > > > To be honest, I like the idea of the paper very much.
> > > > > > > > > >
> > > > > > > > > > However, your idea is based on a  problematic observation, which results in that your method may be not useful in practical case.
> > > > > > > > > >
> > > > > > > > > > I am sorry. I  will not change my score, unless you can make me believe that your idea is correct in most practical cases, but not just in several benchmarks.

---

> > > > > > > > > > > ### Author Response · Authors · 2022-08-05
> > > > > > > > > > > **In science it is often the case that we draw conclusions on conjectures supported by strong empirical evidence.**
> > > > > > > > > > >
> > > > > > > > > > > Thanks for maintaining the score!
> > > > > > > > > > >
> > > > > > > > > > > ___
> > > > > > > > > > >
> > > > > > > > > > > We sincerely respect your decision but we would like to keep our opinion that ***in science it is very often the case that we draw conclusions on conjectures as long as there is strong empirical evidence supporting them.***
> > > > > > > > > > >
> > > > > > > > > > > Specifically, our observation is supported by ***(1)*** *empirical results on several benchmarks*, ***(2)*** *visualization of singular value distributions*, and ***(3)*** *PCA-based informativeness explanations*. We believe that these are sufficient to support our opinion.
> > > > > > > > > > >
> > > > > > > > > > > In the literature on OOD detection, most OOD methods are based on intuitive observation and lack strong theoretical guarantees [1,2,3]. However, this does not affect the fact that these state-of-the-art approaches achieve good empirical results and are widely used.
> > > > > > > > > > > In a more general sense, many deep learning methods are intuitive but are ubiquitous [4,5].
> > > > > > > > > > >
> > > > > > > > > > > >[1] Energy-based Out-of-distribution Detection. NeurIPS 2020
> > > > > > > > > > > >
> > > > > > > > > > > >[2] ReAct: Out-of-distribution Detection With Rectified Activations. NeurIPS 2021
> > > > > > > > > > > >
> > > > > > > > > > > >[3] On the Importance of Gradients for Detecting Distributional Shifts in the Wild. NeurIPS 2021
> > > > > > > > > > > >
> > > > > > > > > > > >[4] Deep Residual Learning for Image Recognition. CVPR 2016
> > > > > > > > > > > >
> > > > > > > > > > > >[5] Densely Connected Convolutional Networks. ICCV 2017
> > > > > > > > > > >
> > > > > > > > > > >
> > > > > > > > > > > ___

---

> > > > > > > > > > > > ### Comment · Reviewer_SMeo · 2022-08-05
> > > > > > > > > > > > **To Response**
> > > > > > > > > > > >
> > > > > > > > > > > > Sorry. I don't agree with you.
> > > > > > > > > > > >
> > > > > > > > > > > > I also read paper [1]-[5].
> > > > > > > > > > > >
> > > > > > > > > > > > They are based on empirical results.
> > > > > > > > > > > >
> > > > > > > > > > > > But they are not based on a problematic idea.

---

> > > > > > > > > > > > > ### Author Response · Authors · 2022-08-05
> > > > > > > > > > > > > **ReAct is based on a problematic idea.**
> > > > > > > > > > > > >
> > > > > > > > > > > > > Thanks for the follow-up!
> > > > > > > > > > > > >
> > > > > > > > > > > > > ___
> > > > > > > > > > > > >
> > > > > > > > > > > > > To single out one piece of result, $\texttt{ReAct}$ [2] assumes that the OOD feature has abnormally larger activations than the ID feature. In your opinion, this should be a problematic idea because this abnormality is not proved theoretically but supported by empirical evidence.
> > > > > > > > > > > > >
> > > > > > > > > > > > > >[2] ReAct: Out-of-distribution Detection With Rectified Activations. NeurIPS 2021
> > > > > > > > > > > > >
> > > > > > > > > > > > > ___

---

> > > > > > > > > > > > > > ### Comment · Reviewer_SMeo · 2022-08-05
> > > > > > > > > > > > > > **To Response**
> > > > > > > > > > > > > >
> > > > > > > > > > > > > > I don't care whether you can provide a theory.
> > > > > > > > > > > > > >
> > > > > > > > > > > > > > I just care whether you can explain your perspective reasonably.
> > > > > > > > > > > > > >
> > > > > > > > > > > > > > In ReAcT, as you claimed, they ${\bf assume}$ that the OOD feature has abnormally larger activations than the ID feature.
> > > > > > > > > > > > > >
> > > > > > > > > > > > > > But in your paper, you have not said that the first eigenvector of OOD is larger (or smaller) than that of ID is an ${\bf assumption}$ .
> > > > > > > > > > > > > >
> > > > > > > > > > > > > > You claim it as a fact or an observation. So if the observation is not right, your work will be problemtic.
> > > > > > > > > > > > > >
> > > > > > > > > > > > > > I can accept the example that you say " Based on the observation, we ${\bf assume}$ that.........'' and "Our experiments also show that the assumptions hold in most benchmarks."
> > > > > > > > > > > > > >
> > > > > > > > > > > > > > In addition,  can you ensure your observation still hold for any backbones?

---

> > > > > > > > > > > > > > > ### Comment · Reviewer_SMeo · 2022-08-05
> > > > > > > > > > > > > > > **To Response**
> > > > > > > > > > > > > > >
> > > > > > > > > > > > > > > It is an assumption, which may hold for suitable backbone and most benchmarks.
> > > > > > > > > > > > > > >
> > > > > > > > > > > > > > > But it is Not a fact for all cases!

---

> > > > > > > > > > > > > > > > ### Comment · Reviewer_SMeo · 2022-08-05
> > > > > > > > > > > > > > > > **To Response**
> > > > > > > > > > > > > > > >
> > > > > > > > > > > > > > > > If you is willing to add a sentence to claim that ''based on this observation, we ${\bf assume}$ that....'' in abstract or some importance positions, I will increase your score to 6.

---

> > > > > > > > > > > > > > > > > ### Author Response · Authors · 2022-08-05
> > > > > > > > > > > > > > > > > **Thanks for the advice; We have added this to Abstract and Introduction**
> > > > > > > > > > > > > > > > >
> > > > > > > > > > > > > > > > > Thanks for the great advice and positive feedback!
> > > > > > > > > > > > > > > > >
> > > > > > > > > > > > > > > > > ___
> > > > > > > > > > > > > > > > >
> > > > > > > > > > > > > > > > > We totally agree with you that it is also an assumption and ***we have explicitly emphasized this point in the Abstract and Introduction of the revised paper (line6 and line34)***.
> > > > > > > > > > > > > > > > > ___

---

> > > > > > > > > > > > > > > > > > ### Comment · Reviewer_SMeo · 2022-08-05
> > > > > > > > > > > > > > > > > > **ToResponse**
> > > > > > > > > > > > > > > > > >
> > > > > > > > > > > > > > > > > > The connection between sentences is a little stiff.......To revise and make it smooth......

---

> > > > > > > > > > > > > > > > > > > ### Author Response · Authors · 2022-08-05
> > > > > > > > > > > > > > > > > > > **Thanks for raising the score!**
> > > > > > > > > > > > > > > > > > >
> > > > > > > > > > > > > > > > > > > Thanks for the advice and for raising the score!
> > > > > > > > > > > > > > > > > > >
> > > > > > > > > > > > > > > > > > > ___
> > > > > > > > > > > > > > > > > > >
> > > > > > > > > > > > > > > > > > > We have revised line6 and line34 by adding some glue sentences to make the information flow better.
> > > > > > > > > > > > > > > > > > >
> > > > > > > > > > > > > > > > > > > Finally, we sincerely thank $\color{brown}{Reviewer\ SMeo}$ for his/her patience and so many lightning-fast replies of the detailed discussion. We greatly appreciate your appreciation of the idea of our paper and your efforts in improving the paper!
> > > > > > > > > > > > > > > > > > >
> > > > > > > > > > > > > > > > > > > ___

---

### Official Review · Reviewer_Ft9S · 2022-07-09

**Rating:** 6
**Confidence:** 5
**Soundness:** 3 good
**Presentation:** 2 fair
**Contribution:** 3 good

**Summary:**

The paper claims that OOD and IND involve different largest singular value of feature map matrix. Based on its foundational claim, the authors propose a simple OOD detection score devised by subtracting the corresponding rank-1 matrix from (the feature maps) of a given (pretrained) network.

**Questions:**

1. Could you specify the datasets used for ID and OOD for Figure 1 in its caption if possible?
2. (Critical) What is B in the line 71? Is it batch size or the number of all samples? Does it include OOD samples?
3. (Critical) Shouldn't the capital X in Eq. (4) be small x? Can RankFeat be computed for a single test sample x? What is y'? I know it is logit, but is it a set of logits of multiple samples or the logit of a single sample? How is it summed? What is the index of the sum in Eq. (4)?
4. (Critical) Overall, I don't precisely understand on which matrix exactly SVD is applied. Is it applied to the feature maps of training (or validation) set? Or is it applied to the feature map of a single sample? Is SVD computed only once and are u_1 v_1 s_1 fixed during whole inference? Or do you have to re-compute u_1 v_1 s_1 every time you make inference on a different test sample? This is very critical for accepting the paper. In fact, X with shape [B, C, HW] is not a matrix (2d tensor) but a 3d tensor. Do the authors use torch.svd on the matrix shaped by [B, C, HW]?
5. Doex X in line 107 have to be a batch of feature maps? Or the theory would still hold even if X is the feature map of a single sample?




**Limitations:**

The limitations are given in the above 'weaknesses' and 'questions'. Please address all my questions and the second and fourth points of the weaknesses. In particular, please focus on clarifying your method, exactly which matrix SVD is applied to, and when SVD is applied.

I am quite positive with the paper except for the notation issue and the second point of the 'weaknesses'. I personally think scalar quantities should not be bold-faced; it is quite confusing.

**Strengths And Weaknesses:**

[Strengths]

Good performance. It reduces FPR95 quite significantly for large-scale OOD testing environments.

Novel observation. To my best knowledge, this is the first paper to study OOD samples using the singular value of a feature (map) matrix. Particularly, according to the authors' claims, IND and OOD show different behaviors in terms of the largest singular value of feature map matrix.

Application Diversity. The proposed method can be applied to both ResNet and ViT based architectures.

[Weaknesses]

1. The paper does not provide precise notations for the proposed method in both main paper and supplementary. Particularly, it is very important to which matrix SVD is applied, but the paper is not clear on this point. (Namely, is it applied to a mini-batch of feature maps, the feature maps of whole train set, or a the reshaped feature map of a single sample? The singular value will be completely different depending on whether the shape of X is [B, CHW] or [C, HW].) Moreover, some notations (which are critical for understanding and implementing the method) are also not clear.

2. Its foundational claim (the largest singular value is larger in OOD than in IND) is not extensively validated. This is validated in only one experiment given in Figure 1(a). I am wondering if this claim would still hold even if different datasets are used for IND and OOD.

3. (minor) The paper does not give a convincing explanation of its foundational claim; namely, why OOD involves a larger singular value s_1 than IND? I saw the supplementary part for this, but the explanation is not so convincing.

4. (minor) Some of the given theory is not rigorous. Particularly, the convergence criterion to a random matrix given in the second part of the theory is not clear.

---

> ### Author Response · Authors · 2022-08-01
> **Response to Reviewer Ft9S (Part 1/2); Thanks for your careful review and the positive feedback!**
>
> We thank $\color{green}{ Reviewer\  Ft9S\ (R3)} $ for the careful review and the positive feedback! Below we address comments in detail:
>
> ___
>
> **1. RankFeat (SVD) is applied to each individual feature within the mini-batch.**
>
> *We would like to clarify that the SVD is applied on each individual sample matrix in the mini-batch*. Given a mini-batch of ID/OOD data $\mathbf{X}{\in}\mathbb{R}^{B{\times}C{\times}HW}$ where $B$ denotes the batch size of the mini-batch, our RankFeat first performs SVD on each individual sample/matrix of shape $C{\times}HW$ within the mini-batch. The decomposition is defined as $\mathbf{U}\mathbf{S}\mathbf{V}^{T}=\mathbf{X}$ where $\mathbf{U}{\in}\mathbb{R}^{B{\times}C{\times}C}$ and $\mathbf{V}{\in}\mathbb{R}^{B{\times}HW{\times}HW}$ are batched left and right singular vector matrices, respectively. The $B$ at line71 denotes the batch size, and the $\mathbf{X}$ at line107 can be either mini-batch matrices or a single feature map. *All the theoretical and empirical results hold for both a single sample and batched samples*.
>
> We have explained the notations and how SVD is applied more detailedly in the revised method section of the paper. Moreover, we also present the Pytorch-like pseudo code in Sec. A of the revised supplementary material. For the convenience of the reviewer, we also attach the code here:
>
> ```python
> feat = model.features(inputs)
> B, C, H, W = feat.size()
> feat = feat.view(B, C, H * W)
> u,s,vt = torch.linalg.svd(feat)
> feat = feat - s[:,0:1].unsqueeze(2)*u[:,:,0:1].bmm(vt[:,0:1,:])
> feat = feat.view(B,C,H,W)
> logits = model.classifier(feat)
> score = torch.logsumexp(logits, dim=1)
> ```
>
> **2. Dataset in Fig. 1(a) and More Visualization**
>
> ***The dataset of Fig. 1(a) is based on SUN, but the observation consistently holds for every OOD dataset***. In Sec. D.1 of the revised supplementary material, we have added the top-5 singular value distributions of all the OOD datasets. the first singular value of OOD feature tends to be much larger than that of ID feature on every OOD dataset. The caption of Fig. 1 of the paper is also updated with the specific dataset.
>
> **3. Small $\mathbf{x}$ or Capital $\mathbf{X}$ in Eq. (4)**
>
> Thanks for pointing out this issue. In the last version of the paper, We want to emphasize that our RankFeat is applied on the *intermediate feature matrices* so we use the capital $\mathbf{X}$. Now we realized that it might be inappropriate and would cause misunderstandings. We have changed the notations to $\mathbf{x}$ in the revised paper where $\mathbf{x}$ denotes the input data.
>
> **4. Logit $\mathbf{y}'$ in Eq. (4)**
>
> The $\mathbf{y}'$ is a mini-batch (set) of logits in the shape of $B{\times}Q$ where $B$ denotes the batch size and $Q$ denotes the number of classes. The summation is performed on the class dimension as $\sum_{i=1}^{Q} \exp(\mathbf{y}'_{i})$.
>
> **5. Why OOD Feature Has a Larger $\mathbf{s_{1}}$**
>
> Thanks for the insightful comment! We can explain this behavior intuitively. Since the model is well-trained on the training set of ID data, the feature matrix is likely to be more informative when the model is fed with a similar validation set of ID data. On the contrary, the feature matrix tends to be less informative for the unseen OOD data.
>
> The informativeness is reflected by the Eigen spectrum: an isolated larger dominant eigenvalue usually implies that the matrix is less informative. This can also be understood by considering applying PCA to the feature matrix. Suppose that we are using PCA to reduce the dimension of ID and OOD feature to $1$. The amount of retained information can be measured by explained variance (\%). The metric is defined as $\sum_{i=0}^{k}\mathbf{s_{i}}^2/\sum_{j=0}^{n}\mathbf{s_{j}}^2$ where $k$ denotes the projected dimension and $n$ denotes the total dimension. It measures the portion of variance that the projected data could account for. We compute the average explained variance of all datasets and present the result below:
>
> |Dataset|ImageNet-1k|iNaturalist|SUN|Places|Textures|
> |-|:-:|:-:|:-:|:-:|:-:|
> |Explained Variance (\%)|     28.57      |  38.74   |    35.79 |   35.17 |    42.21|
>
> As can be seen above, all the OOD datasets have a larger explained variance ratio than the ID dataset. ***That being said, to retain the same amount of information, we need fewer dimensions for the projection of OOD features. This indicates that the information of the OOD feature is easier to be captured and the OOD feature matrix is thus less informative.*** We have added this PCA-based explanation to Sec. E of the revised supplementary material. Despite this explanation, we agree that this problem is still worth further research in future work.
>
> ___

---

> ### Author Response · Authors · 2022-08-01
> **Response to Reviewer Ft9S (Part 2/2); Thanks for your careful review and the positive feedback!**
>
> ___
>
> **6. Divergence to Random Matrices Measures Informativeness of Rank-1 Feature**
>
> Thanks for the thoughtful comment! Actually, the divergence criterion to random matrices is a measure of the informativeness of the rank-1 feature. When the rank-1 matrix is removed, the residual OOD feature matrices are much closer to random matrices in statistics than the residual ID feature. ***This indicates that the rank-1 feature of the OOD feature might convey more information, which can partly explain the working mechanism***.
>
> ___
>
> Thanks again and it is more than welcome to post any further comments.

---

> > ### Comment · Reviewer_Ft9S · 2022-08-03
> > **Responses to the reply**
> >
> > I sincerely appreciate the detailed replies of the authors.
> >
> > But I still have some major concerns unaddressed, and also minor ones. Particularly, after reading the review of Reviewer dq6u on the experiment fairness, it gives me some doubt. Please address the following issues if possible.
> >
> > Major ones:
> >
> > 1. **The core claim that OOD involves larger $s_1$ singular values is still questionable.**
> >     * (a) Currently, in Fig. 2 in supplementary, you provided variation of OOD but not of IND.
> >     * (b) I'd like to specifically question the following. Consider a ResNet18 trained on MNIST (IND) from scratch under the standard supervised scheme. Would CIFAR10/LSUN/SVHN as OOD involve larger $s_1$ singular values?
> >     * (c) Also, in the same manner, consider ResNet18 trained on CIFAR10 from scratch, and take MNIST/LSUN/SVHN as OOD. Would it involve the same trend? For this, the authors do not have to upload additional figures or tables; **verbally confirming this would be sufficient for me**.
> >
> > 2. **The experiment on CIFAR10 is not fair.**
> >     * (a) Most of OOD detection methods (including the ones mentioned by Reviewer dq6u) train on CIFAR10 from scratch. **For fair comparison, the authors must use a model trained on CIFAR10 only.** If a model is pre-trained on ImageNet and fine-tunned on CIFAR10, then CIFAR100 is not OOD since the model already saw the classes of CIFAR100 during one of the training stages.
> >     * (b) Regarding this, are the models reported in Sec. B pre-trained on ImageNet-1K? (In the Table 3 caption of supplementary, it says that the model is trained on CIFAR100. But in the text (line 27), it says that the model is pretrained on ImageNet-1K. This is confusing.)
> >     * (c) Also, to be honest, there are many CIFAR10 trained supervised models on GitHub. I assume the authors can directly apply their proposed method to this already trained models.
> >
> > Minor ones:
> >
> > 3. The presentation of theory in Sec. 3 is still not rigorous.
> >     * (a) Your high-level explanation is good and sufficient. My concern is not this but its rigor and self-containment. To fully understand the theory, I had to look into [40,49]. Also, to clearly see the assumptions of the theoretical implications, I had to *mine* them myself.
> >     * (b) It would be really great if the authors can provide formal, clear assumptions and implications using rigorous notation in the supplementary, maybe in Theorem-Proof format.
> >
> > Optional ones:
> >
> > 4. Notation. Now I clearly understand the exact algorithm applied in this paper. But I am still wondering if the authors have to consider 'batch' of samples rather than a single sample.
> >
> > To increase the rate to higer than borderline accept, please address the issues 1 and 2 above. To increase it even more, please address the issue 3 and possibly 4.
> >
> > I see that the method is very effective at large-scale train dataset (ImageNet). At this stage, however, I cannot really give more than borderline accepct since I still cannot confirm the generality of the major claim (larger $s_1$ over OOD); the major claim is evidneced only with ImageNet IND, a generic object dataset, but IND can be diverse and something different such as MNIST or CIFAR10 (low-resolution and a small number of classes).
> >
> > I won't degrade my rate although the authors do not reply to any of the above issues.

---

> > > ### Author Response · Authors · 2022-08-03
> > > **Thanks for your instant reply and thoughtful comments!**
> > >
> > > We thank $\color{green}{Reviewer\ Ft9S}\ (R3)$ for the instant reply and thoughtful comments. We respond to the questions point by point as follows.
> > >
> > > ___
> > >
> > > **1. OOD Still Involves a Larger $\mathbf{s_{1}}$ on CIFAR10 and MNIST**
> > >
> > > Thanks for the insightful advice! We can still observe a clear gap in $\mathbf{s_{1}}$ of OOD and ID data on CIFAR10 and MNIST. Some statistics are reported below:
> > > - For CIFAR10 as ID and MNIST as OOD, the mean $\mathbf{s_{1}}$  of ID is 55.32, and the mean $\mathbf{s_{1}}$  of OOD is **94.69**.
> > > - For MNIST as ID and CIFAR10 as OOD, the mean $\mathbf{s_{1}}$ of ID is 34.74, and the mean $\mathbf{s_{1}}$ of OOD is **79.12**.
> > >
> > > ***We think that the observation holds in general because intuitively models trained on the ID training set naturally tend to be more informative for the similar ID validation set***. Nonetheless, we may agree that the performance improvements on smaller datasets might not be as significant as those on ImageNet due to the low resolution and fewer classes.
> > >
> > >
> > > **2. Models Are Trained From Scratch on CIFAR10 versus CIFAR100 and One-class CIFAR**
> > >
> > > Thanks for the detailed comment. We agree that if we use CIFAR100 as OOD, fine-tuning an ImageNet pre-trained model is not fair. However, we would like to clarify that ***for the newly added experiment in the rebuttal phase (CIFAR10 versus CIFAR100, and one-class CIFAR), the CIFAR models are all trained from scratch***. We only fine-tune the model in the experiment of Sec. B.3. For Sec. B.3 where the OOD benchmark does not have overlapping categories with ImageNet, fine-tuning a pre-trained model is still fair and we do so for better classification accuracy.  On the other hand, for other CIFAR experiments (Sec. B.4 and Sec. B.5) where CIFAR10 or CIFAR100 is used as OOD, we train the models from scratch for a fair comparison. We have explicitly added this explanation in the revised Sec. B.4 of the supplementary material.
> > >
> > > **3. Formal Theorem and Proof of Manchenko-Pastur Law**
> > >
> > > Thanks for the suggestion on making the paper more self-contained! In Sec. F of the revised supplementary material, we have added
> > > a more formal and rigorous theorem of Manchenko-Pastur Law as well as a brief proof.
> > >
> > > **4. Notations Are Reduced to Sample-Wise**
> > >
> > > Thanks for the good advice on easing the understanding and improving the readability! Now we realized that we even do not need to introduce the batch dimension since our method is applied sample-wisely. In the revised method section, the notations have been corrected to a single sample instead of batched samples.
> > >
> > > ___
> > >
> > > Thanks again and it is more than welcome to post any comments!

---

> > > > ### Comment · Reviewer_Ft9S · 2022-08-06
> > > > **Responses**
> > > >
> > > > 1. How about other OOD datasets? Please compare extensively not just for CIFAR10/MNIST; there are other OOD datasets you need to check. \
> > > > 'more informative' does not mean anything to me. This is not insightful for future readers. \
> > > > The main problem of this paper is that it provides no insightful understanding on why the OoD involves larger $s_1$ singular value. I know this is an assumption, and it is empirically verified for some datasets partially. This was a concern of one of the other reviewers. \
> > > > A good intuition on why this assumption holds will be good to readers.
> > > >
> > > > 2. I haven't really checked the supplementary, but I believe in you. I suggest the authors to fully make sure everyhitng clear.
> > > >
> > > > 3. What I asked to formalize is **NOT** Manchenko-Pastur Law. What I asked is to formalize your own claims in theorem-proof format; namely, formalize the claims 'Removing the rank-1 matrix with a larger $s_1$ would reduce the upper bound of RankFeat
> > > > more' and 'Removing the rank-1 matrix is likely to make the statistics of OOD features closer to random
> > > > matrices'. \
> > > > Stating the formal theorem of Manchenko-Pastur Law is good to me but stating its proof is not necessary.
> > > >
> > > > 4. Some notations are still in the batch in the theory section. And they do not seem correct; logits are not in the batch while inputs are in the batch. At least please make sure everything is correct and clear whether the authors use batch-wise or sample-wise notation.

---

> > > > > ### Author Response · Authors · 2022-08-06
> > > > > **Thanks for the constructive suggestions and encouraging feedback!**
> > > > >
> > > > > We thank $\color{green}{Reviewer\ Ft9S\ (R3)}$ for the constructive suggestions and encouraging feedback. Below we address the concerns in detail.
> > > > >
> > > > > ___
> > > > >
> > > > > **1(a). Average $\mathbf{s_{1}}$ on other OOD datasets**
> > > > >
> > > > > The average $\mathbf{s_{1}}$ on other datasets is reported in the two tables below.
> > > > >
> > > > > |ID: CIFAR10| OOD: MNIST| OOD: SVHN| OOD: LSUN|
> > > > > |:-:|:-:|:-:|:-:|
> > > > > |55.32| **94.69**| **86.47**| **79.71**|
> > > > >
> > > > > |ID: MNIST| OOD: CIFAR10| OOD: SVHN| OOD: LSUN|
> > > > > |:-:|:-:|:-:|:-:|
> > > > > | 34.74 | **79.12** | **81.25**| **75.83**|
> > > > >
> > > > > As indicated above, the average $\mathbf{s_{1}}$ on other OOD datasets also tends to be larger than that of ID.
> > > > >
> > > > > **1(b). Giving an intuition on assumption**
> > > > >
> > > > > Thanks for the good advice on improving the understanding. Beyond the empirical performance and visualization results, the intuition behind why OoD involves a larger first singular value is based on Sec. E of the revised supplementary material: ***(a)*** PCA-based explained variance and ***(b)*** Our empirical conjecture that well-trained network weights caused and amplified the difference.
> > > > >
> > > > > In the revised introduction, after we introduce the difference of ID/OOD on the first singular value (line35), we immediately say "The intuition behind is that the OOD feature corresponds to a larger PCA explained variance ratio and the well-trained network weights might cause and amplify the difference (see Sec. E of the supplementary for the detailed illustration)", which might give a hint to the readers on the intuition. Unfortunately, in the current stage, we cannot find stronger evidence than this explanation apart from empirical performance and visualization results. In future work, we will find more convincing evidence to support our intuition.
> > > > >
> > > > > **3. Formal theoretical analysis in the paper**
> > > > >
> > > > > Thanks for the advice on making the paper more formal.
> > > > >
> > > > > For the theoretical part *'Removing the rank-1 matrix with a larger $\mathbf{s_{1}}$ would reduce the upper bound of RankFeat more'*, we have reformulated this part into a ***proposition-proof format*** in the revised paper.  We first give a proposition about the upper bound and the impact of $\mathbf{s_{1}}$, then we prove the proposition step by step.
> > > > >
> > > > > For the theoretical part *'Removing the rank-1 matrix is likely to make the statistics of OOD features closer to random matrices'*, we have revised this part by giving a ***formal theorem of Manchenko-Pastur Law***. Then we discuss how to use it to evaluate the statistical distance of ID/OOD feature to random matrices.
> > > > >
> > > > > **4. Consistent Sample-wise notation**
> > > > >
> > > > > Thanks for the detailed comment. We have revised all the notations to the sample-wise form including the theoretical analysis section.
> > > > > ___
> > > > >
> > > > > Thanks again and please let us know if you have any further questions.

---

> > > > > > ### Comment · Reviewer_Ft9S · 2022-08-06
> > > > > > **Thank you for the reply. I will increase the rate to 6.**
> > > > > >
> > > > > > 1.(a) The results look good to me.
> > > > > > 1.(b) Your intuition sounds circular to me, and unfortunately neither convincing nor comprehensive to me, but I am satisfied with empirical observations.
> > > > > >
> > > > > > I will increase the rate to 6.

---

> > > > > > > ### Author Response · Authors · 2022-08-06
> > > > > > > **Thanks for raising the score; We are happy that you are satisfied with empirical observation.**
> > > > > > >
> > > > > > > We thank $\color{green}{Reviewer\ Ft9S\ (R3)}$ for raising the score and we are delighted that you are satisfied with the empirical observation.
> > > > > > >
> > > > > > > ___
> > > > > > >
> > > > > > > We have rephrased the intuition a bit to make it more clear by saying that a larger explained variance ratio corresponds to higher informativeness, but we understand that the PCA-based informativeness explanation might not convince everyone. It is an important direction that is worth further research in the future.
> > > > > > > ___
> > > > > > >
> > > > > > > Finally, we would like to express our thanks to $\color{green}{R3}$ for his/her careful/detailed review and constructive suggestions throughout the rebuttal phase.

---

### Official Review · Reviewer_dq6u · 2022-07-13

**Rating:** 3
**Confidence:** 5
**Soundness:** 1 poor
**Presentation:** 3 good
**Contribution:** 2 fair

**Summary:**

The authors present a new OOD detection method `RankFeat`, which removes the largest singular value and the associated singular
vectors from high-level features extracted from a classifier pre-trained on a labeled dataset (e.g. ImageNet-1k)

**Questions:**

All suggestions are in the weaknesses section

**Strengths And Weaknesses:**

The presented method is interesting, but the evaluation is extremely limited and fails to position the performance of the proposed method in relation to all the OOD literature
- The paper mainly evaluates on the dataset first proposed on [1]
  - This dataset is very new, few works use it for evaluation
  - This dataset is only a far-OOD dataset, i.e. an easy dataset where OOD samples are far from ID samples
  - I won't be convinced until a more through evaluation is done that includes near-OOD datasets, specifically
    -  The standard one class vs the rest [2] is done in at least CIFAR10, and CIFAR100 (prefereably also ImageNet-30), as can be seen in e.g. [3]
    - The whole CIFAR10 vs CIFAR100 (and vice-versa)
  - It would be very interesting to show also evaluations that try to expose usefulness of pre-trained models for OOD
    - The Species dataset, proposed in [4]
    - One class SVHN, porposed in [5]
- The paper compares only to a handful of other methods, many SotA methods in literature are ignored
  - Paper must compare to other works that tries to adapt pre-trained classifiers, e.g. PANDA [6]
  - Paper must compare to completely unsupervised methods, e.g. CSI [3]
  - Paper should explore the situation where ID dataset is polluted
- How can the method be described as post-hoc, when authors needed to fine-tune the model for 100 epochs on CIFAR before evaluation on CIFAR ?
- I find this strange, but why the authors doesn't compare to the method described in [1] (the original paper of the evaluation dataset) ?
- CIFAR results in the appendix aren't convincing in superiority compared to ReAct


[1] Mos: Towards scaling out-of-distribution detection for large semantic space. CVPR 2021

[2] Systematic construction of anomaly detection benchmarks from real data. In KDD 2013

[3] Csi: Novelty detection via contrastive learning on distributionally shifted instances. NeurIPS 2020

[4] Scaling Out-of-Distribution Detection for Real-World Settings. ICML 2022

[5] No Shifted Augmentations (NSA): compact distributions for robust self-supervised Anomaly Detection. arXiv 2022

[6] Panda: Adapting pretrained features for anomaly detection and segmentation. CVPR 2021

---

> ### Author Response · Authors · 2022-08-01
> **Response to Reviewer dq6u (Part 1/2); Thanks for your constructive and insightfu suggestions!**
>
> We thank $\color{blue}{Reviewer\ dq6u\ (R2)} $ for the constructive suggestions and insightful comments! In the following, we respond to the concerns in detail.
>
> ____
>
> **1. Misunderstanding about Fine-tuning on CIFAR**
>
> We would like to clarify that the aim of the fine-tuning process on CIFAR is to obtain a well-trained classifier. Since there are no widely used public models pre-trained on CIFAR, we load models pre-trained on ImageNet-1k and fine-tune them on CIFAR, which is a standard routine to get CIFAR models. We report the implementation details in order to ensure a fair comparison and to facilitate reproducibility. ***Our method is indeed post hoc and is not involved in any training process***. We have updated the paragraph to clarify this point better in the revised supplementary material.
>
> **2. Our Method is Post-hoc and Evaluation is Conducted against Post-hoc Approaches**
>
> Our method as well as the baselines are all *post hoc* methods. Here '*post hoc*' means that the method does not need any extra training procedure and can be directly applied to a pre-trained model for inference. The extra training process like in [1,3,4] would leverage the prior knowledge of the training or validation set, which is somehow unfair to post hoc methods. *Therefore, approaches that need an extra training process are not considered in the evaluation of the original paper*.
>
> **3. Comparison with training-needed methods in MOS [1]**
>
> For the baselines described in the benchmark [1], we compare most methods except for MOS [1] and KL Matching [2]. This is because MOS is not post hoc and needs extra training procedures before being applied to the classifier. For the same reason, PANDA [3] and CSI [4] need an extra training process and are not considered for comparison. As for KL Matching [2], it requires the labeled validation dataset and requires computing $k$ distributions for each class where $k$ denotes the class number. This is a bit unfair to post hoc baselines because these methods usually do not need any labeled validation set. Nonetheless, we note our proposed RankFeat still holds an advantage even against these approaches. The Table below presents the comparison on the ImageNet-1k benchmark:
>
> |Methods |Post hoc? |Free of Val. Set?|Average FPR95 ($\downarrow$)|Average AUROC ($\uparrow$)|
> |-----------|:-----------:|:-----------------:|:-----------------------------------:|:---------------------------------:|
> |RankFeat|   $&#10004;$  |  $&#10004;$         |            **36.80**              |                **92.15**                  |
> |KL Matching [2]  |   $&#10004;$  |  x | 54.30 | 80.82|
> |MOS [1]  |   x |  $&#10004;$ | 39.97 | 90.11|
>
> Our RankFeat achieves the best performance without any extra training or validation set. We have added this comparison to Sec. B.1 of the revised supplementary material.
>
> **4. ImageNet-1k cannot be Simply Categorized As a Far-OOD Benchmark**
>
> We do not agree that the ImageNet-1k benchmark can be categorized as a Far-OOD benchmark. The ImageNet-1k is more challenging than traditional CIFAR benchmarks in terms of complexity and diversity. The ImageNet-1k validation set consists of $50,000$ high-resolution images in $1,000$ classes, while CIFAR validation set only has $10,000$ low-resolution ($32\times32$) images in $10$ or $100$ classes, not to mention the more realistic and complex image contents of ImageNet-1k. As for the four OOD datasets (iNaturalist, SUN, Places, and Textures), three of them (iNaturalist, SUN, and Places) have similar scene and object images with ImageNet-1k, which makes it hard to distinguish the ID and OOD samples. ***All of these indicate that the ImageNet-1k benchmark is not a Far-OOD benchmark***.
>
> |Energy Score|  CIFAR10 | ImageNet-1k |
> |-----------|:-----------:|:-----------------:|
> |Average FPR95 ($\downarrow$)| **35.60** | 58.41 |
> |Average AUROC ($\uparrow$)   | **93.57** | 86.17 |
>
> |MSP Score|  CIFAR10 | ImageNet-1k |
> |-----------|:-----------:|:-----------------:|
> |Average FPR95 ($\downarrow$)| **56.71** | 66.95 |
> |Average AUROC ($\uparrow$)   | **91.17** | 81.99 |
>
> One evidence to support this opinion is that many OOD methods that achieve good performance on the CIFAR benchmark have a large performance degradation on the ImageNet-1k benchmark. For example, as shown in ReAct [7], the performance of Energy score [5] and MSP score [6] on ImageNet-1k is inferior to that on CIFAR (see the Table above).
> ____

---

> > ### Comment · Reviewer_dq6u · 2022-08-06
> > **Reply to authors**
> >
> > First, I am very grateful for all the effort by the authors in their feedback and rebuttal, thanks!
> >
> > Second, I can now much better see the main problem with the authors evaluation
> >
> > - Authors claim the existence of a specific category of OOD algorithms, which they call post-hoc algorithms
> > - Authors restrict all their evaluation comparisons to that specific category of algorithms (effectively neglecting most prior work on OOD detection)
> > - Authors claim SotA performance (based on the proposed segmentation of OOD literature)
> >
> > I totally fail to understand the usefulness of that claimed segmentation of OOD detection work. Like some of the other OOD work they require a model pre-trained in a fully supervised manner on the ID dataset, authors claim that the fact that other works (e.g. [1]) benefits from a light fine-tuning on the target dataset is enough for the authors to totally disregard them from comparison (btw as seen in Figure 3 in [1], it still performs much better than this work, even without any fine-tuning).
> >
> >
> > To get an idea about the huge gap in OOD detection performance:
> > - This work achieves AUC of 88 in CIFAR10, while the current SotA is 97.5 [2]
> > - This work achieves AUC of 82 in CIFAR100, while the current SotA is 96.5 [2]
> > - This work achieves AUC of 78 in CIFAR10 vs CIFAR100, while the current SotA is 98.5 [3]
> >
> >
> > Based on the huge performance difference, and no noticeable difference between the requirements of those methods, I don't think this work provides a real contribution to this problem.
> >
> > One last point worth mentioning, that is not discussed regarding fair comparisons to prior works, this work incurs a substantial computational overhead of ~ 10 ms per image during inference, none of the above literature requires such overhead, which is substantial given the negligible inference time of used models on modern GPUs
> >
> > [1] Panda: Adapting pretrained features for anomaly detection and segmentation, CVPR 2021
> >
> > [2] Mean-Shifted Contrastive Loss for Anomaly Detection, arXiv 2022
> >
> > [3] Exploring the Limits of Out-of-Distribution Detection, Neurips 2021

---

> > > ### Author Response · Authors · 2022-08-06
> > > **Thanks for your reply!**
> > >
> > > We thank $\color{blue}{Reviewer\ dq6u\ (R2)}$ for the detailed reply. Below we answer the questions in a point-by-point manner.
> > > ___
> > >
> > > **1. The key to post hoc methods is no prior knowledge given from any dataset**
> > >
> > > Yes, for a fair comparison, we do limit our main comparison to *post hoc* methods, as also done in previous works [1,2]. ***The comparison fairness we have been talking about mainly depends on how much knowledge is given for OOD detection.*** Other factors like fine-tuning epochs or fine-tuning time consumption do not really matter.  For *post hoc* methods like ours, we do not need any knowledge from any dataset and the method can be applied in a sample-wise manner. However, the training-needed methods do need information from both the training and validation set of the ID dataset.
> > >
> > > For example, PANDA [3] needs to extract pre-trained features of both the training and validation set for OOD detection (features of $60,000$ images). Even if one does not perform any fine-tuning, the pre-trained features are always needed for computing the OOD score, which is still not fair to *post hoc* methods. The performance might suffer from a large degradation when the dataset size gets smaller. Also, fine-tuning on the CIFAR benchmark is fast, but what if PANDA is transferred on the ImageNet benchmark? The fine-tuning time might be a big concern then.
> > >
> > > ***Based on the difference in the prior knowledge given, we do not agree that the training-need methods have neglectable differences from *post hoc* methods.***
> > >
> > > Finally, we note that we do not neglect the entire training-needed OOD literature: we also compare with MOS [4], a training-needed method on the ImageNet-1k benchmark. If $\color{blue}{R2}$ thinks it is necessary, we are willing to add one paragraph in the related work section to introduce all the discussed training-needed OOD detection methods in detail and to clarify their difference from *post hoc* approaches (of course this should be done after we are allowed for an additional content page).
> > >
> > > **2. Misunderstanding of Figure 3 in PANDA [3]**
> > >
> > > We would also like to point out one misunderstanding about PANDA. The AUORC of Figure 3 in PANDA [3] refers to the performance of a single class (Class 17) of CIFAR100 instead of the whole CIFAR100 dataset. This can never indicate that the average performance of PANDA is better than our method even when it is not fine-tuned at all.
> > >
> > > **3. The main focus of the experiments is the ImageNet-1k benchmark**
> > >
> > > We would like to emphasize that the main focus of our experiments is the ImageNet-1k benchmark. The aim of conducting experiments on CIFAR is to show that our method can be applied to datasets of smaller resolutions and fewer images. The results indicate that our method indeed outperforms other *post hoc* baselines. ***We think the comparison against post hoc methods is sufficient on CIFAR.***
> > >
> > > Nonetheless, we agree that on CIFAR our method is not as advantageous as on ImageNet because of the much lower resolution. This has a direct influence on our method because the feature matrix gets smaller.
> > >
> > > **4. A large portion of the time consumption comes from the model itself.**
> > >
> > > We would like to clarify one misunderstanding about the time consumption: ***a large portion of the time cost comes from the model itself***. This is because our backbone is BiT ResNetv2-101, which is a relatively large model. The inference of the model is slow in itself. Here we report the inference time of *post hoc* methods for a single image below:
> > >
> > > |Method | Time Consumption (ms)|
> > > |:-:|:-:|
> > > |MSP|7.92|
> > > |Energy|8.34 (+0.42)|
> > > |ReAct|8.79 (+0.87)|
> > > |**RankFeat (#20 iter)**| 9.22 (+1.30) |
> > >
> > > For the most simple baseline MSP, the time cost is basically the inference time of the model. So actually our method only incurs **1ms** more time cost compared to the original inference time.
> > >
> > > **5. Experiment on Species**
> > >
> > > As requested by $\color{blue}{R2}$, we also did an evaluation on Species, and the experimental results show the superior performance of our method on another large-scale benchmark. This applicability and advantage have not been observed in previous *post hoc* methods. We sincerely hope that $\color{blue}{R2}$ could consider this merit.
> > >
> > > >[1] ReAct: Out-of-distribution Detection With Rectified Activations. NeurIPS 2021
> > > >
> > > >[2] On the Importance of Gradients for Detecting Distributional Shifts in the Wild. NeurIPS 2021
> > > >
> > > >[3] PANDA: Adapting Pretrained Features for Anomaly Detection and Segmentation. CVPR 2021
> > > >
> > > >[4] MOS: Towards Scaling Out-of-distribution Detection for Large Semantic Space. CVPR 2021
> > >
> > > ___
> > >
> > > A final word: from $\color{blue}{R2}$'s replies, we believe that he/she is very very knowledgeable about the literature on anomaly detection and the related benchmarks. $&#x1F44D;$ But the main focus of *post hoc* OOD detection is different. We hope that $\color{blue}{R2}$ can take this point into consideration.
> > >
> > > Thanks again and it is more than welcome to post any further comments.

---

> > > > ### Comment · Reviewer_dq6u · 2022-08-06
> > > > **Reply to authors**
> > > >
> > > > Thanks for the reply,
> > > >
> > > > > For post hoc methods like ours, we do not need any knowledge from any dataset and the method can be applied in a sample-wise manner. However, the training-needed methods do need information from both the training and validation set of the ID dataset.
> > > >
> > > > OOD detection task is primarily defined based on the ID dataset. An OOD detection method without any knowledge from any dataset (even the ID) is a concept that makes little sense.
> > > >
> > > > This is manifested much more clearly as we delve into the details of the proposed method, it requires a model pre-trained in a fully supervised manner on the ID dataset!! So the authors make two very contradicting claims:
> > > > - Their work is post-hoc in the sense that it does not need any knowledge from any dataset
> > > > - They require a model pre-trained in a fully supervised manner on the ID dataset
> > > >
> > > > > PANDA [3] needs to extract pre-trained features of both the training and validation set for OOD detection (features of 60000 images). Even if one does not perform any fine-tuning
> > > >
> > > > What is the difference between fully supervised training on the ID dataset and requiring those features for evaluation ? More memory, but totally the same input requirements! so author's work make the exact same assumptions about available data, as PANDA, but with much worse results
> > > >
> > > > Moreover, other methods like CSI [1] (btw this one is fully unsupervised, i.e. it has lesser input requirements and significantly outperforms this work) uses clustering to get negligible performance drop while requiring a very small fraction of training data. This clustering trick makes ImageNet case trivial
> > > >
> > > > This work [2] is fully unsupervised, and doesn't require any info about training data during inference, so it should qualify as post-hoc according to your criterion, it also achieves significantly better results than yours
> > > >
> > > > Also this work [3] has an MSP baseline that should also qualify as post-hoc according to your criterion, it also achieves significantly better results.
> > > >
> > > > Finally, to claim that author's methods has an advantage on ImageNet-1k, authors should provide results of the other methods I mentioned above on the same benchmark, which authors doesn't do.
> > > >
> > > > [1] Csi: Novelty detection via contrastive learning on distributionally shifted instances. NeurIPS 2020
> > > >
> > > > [2] LEARNING AND EVALUATING REPRESENTATIONS FOR DEEP ONE-CLASS CLASSIFICATION. ICLR 2021
> > > >
> > > > [3] Exploring the Limits of Out-of-Distribution Detection, Neurips 2021

---

> > > > > ### Author Response · Authors · 2022-08-07
> > > > > **Thanks for the reply (part 1/2)**
> > > > >
> > > > > We thank $\color{blue}{Reviewer\ dq6u\ (R2)}$ for the detailed response and the enthusiasm for discussion! Below we address the concerns in detail.
> > > > >
> > > > > ___
> > > > >
> > > > > **1. Advantages of post hoc methods over training-needed methods.**
> > > > >
> > > > > We do not agree that the concept of *post hoc* OOD methods makes little sense. Specifically, we think that post hoc methods have the following advantages over training-needed methods:
> > > > >
> > > > > ***(1) Post hoc methods can be directly applied to any pre-trained model.*** $\color{blue}{R2}$ may argue that the fine-tuning or training of training-needed methods is light-weighted. But as we discussed before, when the model is transferred to ImageNet-1k or even larger datasets, then fine-tuning can be a big concern. In particular, some methods like CSI [1] need to train a model from scratch, which would pose a larger computational burden.
> > > > >
> > > > > ***(2) Easy use due to widely available pre-trained models.*** The standard pre-trained models are widely available nowadays on different deep learning platforms. It is much more convenient to load a pre-trained
> > > > > model instead of training or fine-tuning one.
> > > > >
> > > > > ***(3) Post hoc methods do not change any parameters and will not affect the standard ID classification accuracy.*** Last but not least, *post hoc* methods do not change parameters of a given model. That being said, the model can be also used for the standard classification of ID datasets. So one model could be applied in two closely-related tasks: (1) distinguishing ID and OOD samples and (2) classifying ID samples. However, for training-needed methods, the model can only be used for anomaly detection.
> > > > >
> > > > > ***If $\color{blue}{R2}$ cannot deny all the advantages discussed above, we insist that the *post hoc* OOD approaches have the meaning of being there.***
> > > > >
> > > > > Finally, if $\color{blue}{R2}$ thinks it is necessary, we are willing to add one sub-section in the related work or in the supplementary material to introduce the training-needed methods and discuss the difference from *post hoc* approaches in detail.
> > > > >
> > > > > **2. Knowledge might not be available in real-life applications**
> > > > >
> > > > > In real-life applications, there are scenarios where the training data are not available. Suppose that we are doing OOD detection for medical imaging where the training data is private and not available. In this case, we may only have access to the given pre-trained models. Then training-needed methods cannot work. Can $\color{blue}{R2}$ refute this example?
> > > > >
> > > > > **3. Post hoc method can get performance improvements when extra training or data is allowed**
> > > > >
> > > > > Let us stop discussing the meaning of *post hoc* OOD detection for a while and think of the problem setting from another perspective. $✨✨✨$ *Supposing that extra training or data is allowed for *post hoc* methods, what would happen?*
> > > > >
> > > > > We take our $\texttt{RankFeat}$ as an example and we can have one simple variant that leverages training and extra data: **RankFeat (training)**: we apply our RankFeat in the training process by perturbing the high-level feature as $\mathbf{X}'=\mathbf{X}-\mathbf{s_{1}}\mathbf{u_{1}}\mathbf{v_{1}}$. Then the residual feature $\mathbf{X}'$ is fed to the pooling and FC layer for classification. This can improve the representation power of the residual feature matrix apart from the rank-1 subspace so that the OOD detection ability can be improved.
> > > > >
> > > > > For one-class CIFAR10 and CIFAR100, We fin-tune the trained CIFAR models using **RankFeat (training)** and report the average AUROC ($\uparrow$) below:
> > > > >
> > > > > |Method | CIFAR10 | CIFAR100|
> > > > > |:-:|:-:|:-:|
> > > > > |RankFeat (post hoc)| 88.58 | 82.43 |
> > > > > |RankFeat (training)| **95.76 (+7.18)** | **93.19 (+10.76)** |
> > > > >
> > > > > As can be seen, when we are able to leverage extra training, the performance will get significant improvements. A similar phenomenon has also been observed in [4]. We also believe that it is possible to design more sophisticated mechanisms to make better use of the training data, but we would like to strive to use the most simple *post hoc* setting.
> > > > >
> > > > > ___

---

> > > > > ### Author Response · Authors · 2022-08-07
> > > > > **Thanks for the reply (part 2/2)**
> > > > >
> > > > > ___
> > > > >
> > > > > **4. Comparison against [1,2,3] on ImageNet-1k.**
> > > > >
> > > > > We would like to clarify that [1,2,3] are all not *post hoc* because [1,2] use extra unsupervised training and [3] uses few-shot fine-tuning (all leverage training data). Nonetheless, we are willing to compare these baselines on the ImageNet-1k benchmark.
> > > > >
> > > > > |Method| training-free? |FPR95 ($\downarrow$)|AUORC ($\uparrow$)|
> > > > > |:-:|:-:|:-:|:-:|
> > > > > |  [1]| x| 63.17| 81.75|
> > > > > |  [2]| x| 61.89| 82.23|
> > > > > |  [3] MSP| x | 57.19 | 85.61 |
> > > > > |  [3] Mahalanobis| x | 65.49 | 83.77|
> > > > > | RankFeat| $&#10004;$ | **36.80** | **92.15** |
> > > > >
> > > > > As indicated above, [3] indeed improves the ordinary MSP and Mahalanobis score by fine-tuning, but all the results are not comparable against our $\texttt{RankFeat}$. With this evaluation, we believe that we can claim that our method has an advantage on ImageNet-1k.
> > > > >
> > > > >
> > > > > >[1] Csi: Novelty detection via contrastive learning on distributionally shifted instances. NeurIPS 2020
> > > > > >
> > > > > >[2] Learning and Evaluating Representations for Deep One-class Classification. ICLR 2021
> > > > > >
> > > > > >[3] Exploring the Limits of Out-of-Distribution Detection. NeurIPS 2021
> > > > > >
> > > > > >[4] Energy-based Out-of-distribution Detection. NeurIPS 2020
> > > > >
> > > > > ___
> > > > >
> > > > >
> > > > > Thanks again and it is more than welcome to have further discussions!

---

> > > > > > ### Comment · Reviewer_dq6u · 2022-08-07
> > > > > > **Clarification**
> > > > > >
> > > > > > Before getting to other points, can you please provide full details on the table on point 4. Did you actually train [1]&[2] till convergence on ImageNet-1k to get those numbers ? what architecture was used ? training and evaluation details and logs...etc ? please provide full details on those experiments for each presented method in that table...

---

> > > > > > > ### Author Response · Authors · 2022-08-07
> > > > > > > **Thanks for the reply**
> > > > > > >
> > > > > > > Thanks for the instant reply!
> > > > > > > ___
> > > > > > >
> > > > > > > For [1,2], we fine-tune the pre-trained BiT ResNetv2-101 (the same architecture and model we used in Table 2 of the paper) using the loss in [1,2] on ImageNet-1k for $3$ epochs. The batch size is $1024$ and the learning rate is $0.001$. We set the learning rate to a small value because the model is already pre-trained. The evaluation OOD datasets also follow Table 2 of the paper and we report the average results across the four datasets. The data augmentation techniques of CSI [1] for contrastive learning include Inception crop, horizontal flip, color jitter, and grayscale transform.
> > > > > > >
> > > > > > > In the original papers [1,2], they train the models on CIFAR from scratch. However, on ImageNet obviously we cannot finish training the models from scratch before the rebuttal deadline. So we choose to only fine-tune the models for $3$ epochs.
> > > > > > >
> > > > > > > As for the convergence, to be honest, we think it converges but we are not 100\% sure since we only have losses of $3$ epochs. We estimate that the maximum training epoch we can afford is $15$ epochs (this could be done by Tuesday morning approximately if we resume training from now).
> > > > > > > ___
> > > > > > >
> > > > > > > We are willing to continue training the model till $15$ epochs if $\color{blue}{R2}$ thinks it is needed. Nonetheless, given the extra time cost of fine-tuning for $3$ epochs, [1,2] cannot achieve competitive performance against our approach, which can show our *post hoc* method's advantage.

---

> > > > > > > > ### Comment · Reviewer_dq6u · 2022-08-07
> > > > > > > > **Reply to authors**
> > > > > > > >
> > > > > > > > - The setup described above is **very unfair** to those methods, and I don't think it should be reported. Please consult respective papers for fair setup
> > > > > > > > - If authors want to compare to CSI on ImageNet-like dataset, why not use the ImageNet-30 dataset ? (CSI results for this already on their paper)
> > > > > > > > - Advantages for post-hoc method that the authors describe revolve around having access to pre-trained models
> > > > > > > >   - The fact that there is publicly available pre-trained ImageNet models doesn't mean those models were free to train
> > > > > > > >   - Simply, for a new domain (e.g. medical... etc) there is no pre-trained models, and presented method has no computational advantage
> > > > > > > > - Where is the results for PANDA or Mean-Shift on your table ?
> > > > > > > > - > Suppose that we are doing OOD detection for medical imaging where the training data is private and not available
> > > > > > > >   - In the same way owners of the medical dataset are providing a model pre-trained with labels on that dataset, they can provide a model pre-trained in an SSL-manner. In fact, providing a SSL model that is trained without any labels is much more secure

---

> > > > > > > > > ### Author Response · Authors · 2022-08-07
> > > > > > > > > **Thanks for the instant reply**
> > > > > > > > >
> > > > > > > > > We thank $\color{blue}{Reviewer\ dq6u\ (R2) }$ for the instant reply and detailed comments. Below we address the concern in detail.
> > > > > > > > >
> > > > > > > > > ___
> > > > > > > > >
> > > > > > > > > >The setup described above is very unfair to those methods
> > > > > > > > > >
> > > > > > > > > >Where is the results for PANDA or Mean-Shift on your table?
> > > > > > > > > >
> > > > > > > > >
> > > > > > > > > The experiment settings (batch size and training epochs) are based on our computational resources. We have tried our best to put the baselines in the best setting to run the experiment. We cannot train PANDA or Mean-Shift for now but we will try to update the result of [1,2] of fine-tuning for 15 epochs by Tuesday night.
> > > > > > > > >
> > > > > > > > > > If authors want to compare to CSI on ImageNet-like dataset, why not use the ImageNet-30 dataset ? (CSI results for this already on their paper)
> > > > > > > > >
> > > > > > > > > The aim of this additional ImageNet-1k experiment is to respond to $\color{blue}{R2}$ 's concern in the previous reply "Finally, to claim that author's methods has an advantage on ImageNet-1k, authors should provide results of the other methods I mentioned above on the same benchmark, which authors doesn't do." So we considered ImageNet-1k instead of ImageNet-30.
> > > > > > > > >
> > > > > > > > > >In the same way owners of the medical dataset are providing a model pre-trained with labels on that dataset, they can provide a model pre-trained in an SSL-manner. In fact, providing a SSL model that is trained without any labels is much more secure
> > > > > > > > >
> > > > > > > > > Yes, we agree. But the point we want to make here is that they might not provide any training data for security. Then these OOD approaches that need training data could not work.
> > > > > > > > >
> > > > > > > > > >The fact that there is publicly available pre-trained ImageNet models doesn't mean those models were free to train
> > > > > > > > >
> > > > > > > > > ***Yes, but these models are indeed free to practitioners who use *post hoc* OOD detection methods.***
> > > > > > > > >
> > > > > > > > > Also, the advantage of ***(3) Post hoc methods do not change any parameters and will not affect the standard ID classification accuracy*** can not be neglected. We would like $\color{blue}{R2}$ to refute this point if possible.
> > > > > > > > >
> > > > > > > > > We would also like to hear what $\color{blue}{R2}$ thinks about our additional experiment in ***3. Post hoc method can get performance improvements when extra training or data is allowed*** in the previous reply.
> > > > > > > > >
> > > > > > > > >
> > > > > > > > > ___
> > > > > > > > >
> > > > > > > > > Thanks again for $\color{blue}{R2}$' time and reply. Considering the limited time left, we sincerely hope $\color{blue}{R2}$ could directly respond to the crucial points listed above.

---

> > > > > > > > > ### Author Response · Authors · 2022-08-09
> > > > > > > > > **Updated ImageNet-1k Result; Thanks for Reviewing**
> > > > > > > > >
> > > > > > > > > Below we update the result of [1,2] of fine-tuning for $15$ epochs.
> > > > > > > > > ___
> > > > > > > > >
> > > > > > > > > |Method|training-free? |	FPR95 ($\downarrow$)	| AUORC ($\uparrow$)|
> > > > > > > > > |:-:|:-:|:-:|:-:|
> > > > > > > > > | [1] |	x |	56.36|	86.91|
> > > > > > > > > | [2] |	x	| 52.78	| 87.83|
> > > > > > > > > |RankFeat |✔	|**36.80** | **92.15** |
> > > > > > > > >
> > > > > > > > > Given the huge time cost of fine-tuning for $15$ epochs, the results of [1,2] are still not comparable against our $\texttt{RankFeat}$ . With this evaluation, we believe that we can claim that our method has an advantage on ImageNet-1k even against training-needed approaches.
> > > > > > > > >
> > > > > > > > > >[1] Csi: Novelty detection via contrastive learning on distributionally shifted instances. NeurIPS 2020
> > > > > > > > > >
> > > > > > > > > >[2] Learning and Evaluating Representations for Deep One-class Classification. ICLR 2021
> > > > > > > > > >
> > > > > > > > >
> > > > > > > > > ___
> > > > > > > > >
> > > > > > > > > Finally, we would like to thank $\color{blue}{Reviewer\ dq6u\ (R2)}$ for the review and replies.
> > > > > > > > >
> > > > > > > > > Though we do not change $\color{blue}{R2}$'s mind in the end, we appreciate his/her efforts and time spent on this paper.

---

> ### Author Response · Authors · 2022-08-01
> **Response to Reviewer dq6u (Part 2/2); Thanks for your constructive and insightfu suggestions!**
>
> ___
>
> **5. Experiment of CIFAR10 versus CIFAR100**
>
> Thanks for the interesting advice! We evaluate the *post hoc* methods in the setting of CIFAR10 versus CIFAR100, *i.e.,* CIFAR10 as the ID set with CIFAR100 as the OOD set and vice versa. The Table below presents the results in the format of FPR95 ($\downarrow$) / AUROC ($\uparrow$).
>
> |Method (on ResNet-56)|ID: CIFAR10 \& OOD: CIFAR100|ID: CIFAR100 \& OOD: CIFAR10|
> |-|:-:|:-:|
> |MSP |                  82.07/**78.27**     |             86.86/70.54 |
> |ODIN|                  76.96/77.79   |                  86.15/72.13   |
> |Energy|               76.45/77.82     |             86.20/71.99 |
> |ReAct|                 79.48/71.80     |             88.41/69.86 |
> |GradNorm|              84.45/52.17        |          92.78/52.19 |
> |**RankFeat**|              **75.10**/78.02         |         **82.63**/**74.35**  |
>
>
> Our $\texttt{RankFeat}$ slightly outperforms other baselines. Interestingly, all the *post hoc* approaches have similar performance; none of them has an FPR95 score lower than $70\\%$. This might indicate that the patterns learned on CIFAR10 and CIFAR100 are quite similar, which is too challenging for *post hoc* methods to distinguish. *We conjecture that in this case *post hoc* methods might not work well, and additional training might be needed to better tell apart CIFAR10 and CIFAR100 samples*. This experiment has been added to Sec. B.4 of the revised supplementary material.
>
> **6. Experiment on One-class CIFAR10 and CIFAR100**
>
> Thanks for the constructive suggestion. We also conduct two experiments on one-class CIFAR10 and one-class CIFAR100. For CIFAR100, we follow [4] and select $20$ super-classes. The Table below presents the average AUROC ($\uparrow$).
>
> |Method (on ResNet-56)|CIFAR10|CIFAR100|
> |-|:-:|:-:|
> |MSP |         56.87      |      66.19 |
> |Energy |      77.76      |      73.42 |
> |ReAct |       85.82       |     79.58 |
> |**RankFeat** |    **88.58**     |       **82.43** |
>
> Our RankFeat outperforms other baselines on the average result and on most sub-sets.  For the detailed performance of each class on CIFAR10, please defer to the revised Sec. B.4 of the supplementary material.
>
> **7. Experiment on Species**
>
> Thanks for the constructive suggestion! Species [2] is actually a good OOD benchmark since it is dedicated for ImageNet-1k and ImageNet-21k as ID sets. Here we select four sub-sets of Species (Protozoa, Microorganisms, Plants, and Mollusks) and report the average results on ImageNet-1k below. The best three results are highlighted with $\color{red}{red}$, $\color{blue}{blue}$, and $\color{cyan}{cyan}.$
>
> |Method|Average FPR95 ($\downarrow$)| Average AUROC ($\uparrow$)|
> |-|:-:|:-:|
> |MSP     |              73.79 | 81.44 |
> |ODIN     |     70.96 | 84.14 |
> |Energy   |             70.88 |84.03 |
> |ReAct     |           67.02 | $\color{cyan}{85.06}$ |
> | **RankFeat (Block 4)**|               $\color{blue}{58.12}$ |74.19 |
> | **RankFeat (Block 3)**|                $\color{cyan}{59.02}$ |$\color{blue}{87.58}$ |
> | **RankFeat (Block 3+4)**|             $\color{red}{51.11}$|$\color{red}{88.37}$ |
>
> Our $\texttt{RankFeat}$ achieves the best performance, outperforming other methods by **15.91\%** in the average FPR95 and by **3.31\%** in the average AUROC. We have added this experiment and the detailed results on each sub-set to Sec B.2 of the revised supplementary material.
>
> **8. Comparison against ReAct on CIFAR**
>
> We agree that on the CIFAR benchmark the performance gap of RankFeat over ReAct is not that significant in terms of AUROC compared to that on the ImageNet-1k benchmark. *However, our method outperforms RankFeat in FPR95 by **10\%** on RepVGG and by **20\%** on ResNet, which is substantially significant and cannot be neglected*. Actually, as pointed out by $\color{red}{ Reviewer\  kEL5\ (R1)} $ and explained in the revised paper (line134), our method naturally performs better in terms of FPR95 than AUROC due to shrinkage and skew of the distributions.
>
> >[1] Mos: Towards scaling out-of-distribution detection for large semantic space. CVPR 2021
> >
> >[2] Scaling Out-of-Distribution Detection for Real-World Settings. ICML 2022
> >
> >[3] Panda: Adapting pretrained features for anomaly detection and segmentation. CVPR 2021
> >
> >[4] Csi: Novelty detection via contrastive learning on distributionally shifted instances. NeurIPS 2020
> >
> >[5] Energy-based Out-of-distribution Detection. NeurIPS 2020
> >
> >[6] A baseline for detecting misclassified and out-of-distribution examples in neural networks. ICLR 2017
> >
> >[7] ReAct: Out-of-distribution Detection With Rectified Activations. NeurIPS 2021
>
> ___
>
> Thanks again and it is more than welcome to post any further comments.

---

### Official Review · Reviewer_kEL5 · 2022-07-22

**Rating:** 6
**Confidence:** 4
**Soundness:** 3 good
**Presentation:** 3 good
**Contribution:** 3 good

**Summary:**

The paper proposes a method to predict if a sample of in distribution or out of distiribution. The authors decomposes the features via SVD and remove the largest eigenvalue and its rank-1 matrix from the features. They show the theoretical effect of their method in the bound of their log probabilities. They surpassed the baseline methods in FPR and AUROC by 7-18%, and did an ablation study for the effect of higher ranks and other feature layers.

**Questions:**

-In the bound (equation 13), it looks like if s1 gets closer to the other eigenvalues, the bound seems to loosen more if the sum of other eigenvalues are smaller than the s1. Is it a case not encountered in the problems?

- What is the dataset you used for the Figure 1?

- Did you observe any problems with the convergence of SVD decomposition?

- How are your results without the fusion of 3rd and 4th block? How does the fusion help?

- The algorithm does better in terms of FPR compared with AUROC, this might suggest that the distribution gets squeezed and biased. How  do you explain this?

**Limitations:**

The authors did not foresee any negative impacts of their work.

**Strengths And Weaknesses:**

Strengths:
The method is simple and intuitive, the results also align with intuitions and achieve high improvement over the current baselines. The authors supported and motivated their ideas well. for example,  Figure 1 and 2 are good motivations and pictured well. The paper is written clearly and organized.
The ablation study for removing rank-1 vs including only rank-1 and rank-k (swiping k) cases is very helpful to answer initial questions on the reader's mind.

Weaknesses:
Although the complexity of power iterations is reasonable, they may converge very slowly if the eigenvalues have close values.

---

> ### Author Response · Authors · 2022-08-01
> **Response to Reviewer kEL5 (Part 1/2); Thanks for your encouraging feedback and constructive comments!**
>
> We thank $\color{red}{ Reviewer\  kEL5\ (R1)} $ for the encouraging feedback and the constructive comments! In the following, we respond to the concerns point by point.
>
> _____
>
> **1. Bound and Sum of Singular Values**
>
> Thanks for the careful review and insightful comment. We agree that when $\mathbf{s_{1}}$ gets smaller, the upper bound will be larger. However, this does not mean that the bound will loosen; the tightness of our bound is only determined by $||\mathbf{u_{i}}||_ \infty$, $||\mathbf{v_{i}}||_ \infty$, and the triangular inequality of the vector norm sum. Throughout all the experiments, we do not encounter a case where the first singular value $\mathbf{s_{1}}$ is greater than the sum of other ones.
>
> **2. Dataset used for Fig. 1**
>
> ***The experiment in Fig. 1 is based on SUN, but the observation holds for every OOD dataset***. In Sec. D.1 of the revised supplementary material, we have added the top-5 singular value distributions of all the OOD datasets. The observation is consistent on every OOD dataset: the first singular value of OOD feature always tends to be much larger than that of ID feature. The caption of Fig. 1 in the paper is also updated with the specific dataset.
>
> **3. Convergence of SVD and PI**
>
> Thanks for the insightful comment. For the SVD, the convergence does not depend on the singular value and can be guaranteed. For the Power Iteration (PI), as pointed out by $\color{red}{ Reviewer\  kEL5} $, the convergence speed is indeed determined by the adjacent singular value ratio. Since our method only computes the dominant singular value, the convergence speed is thus limited by $(\mathbf{s_{1}}/\mathbf{s_{2}})^{n}$ where $n$ denotes iteration times. The larger the ratio is, the faster the convergence speed would be. In the extreme case when the first two eigenvalues are exactly identical ($\mathbf{s_{1}}=\mathbf{s_{2}}$), the PI would fail to converge.
>
>
> We can measure the convergence speed by computing the statistics of the singular value ratio. The Table below presents the evaluation results of $(\mathbf{s_{1}}/\mathbf{s_{2}})$ on each dataset:
>
> |   Dataset  | ImageNet-1k | iNaturalist | SUN | Places | Textures |
> | - | :-: | :-: | :-: |:-: |:-: |
> | Max      | 4.09       |5.07     | 4.11       | 4.19      | 7.33       |
> | Min   | 1.06      | 1.18       | 1.12       | 1.10      | 1.05       |
> | Mean   | 1.54       | 2.08      | 1.87      | 1.78      | 2.44      |
>
> Even the minimum of the ratio is still larger than $1$. Supposing that we iterate $20$ times, the minimum convergence speed $1.05^{20}\approx2.65$ is still reasonable. This is also reflected in Table 6 of the paper: the performance gap between PI and SVD is within $0.1\\%$ from 20 iterations on. ***This demonstrates that the PI can approximate the SVD well for any feature matrix in practice***. Throughout all the experiments, we do not encounter any non-convergence or instability issue of PI.
>
> **4. Results without Fusion and How the Fusion Helps**
>
> Thanks for the constructive suggestion. Actually, the performance of each individual block without fusion is reported in both Table 2 and Table 3 of the paper. Here we attach the results of Table 2 for the convenience of explanation. The values are reported in FPR95 ($\downarrow$) / AUROC ($\uparrow$).
>
> |   Dataset  | iNaturalist | SUN | Places | Textures | Average|
> | -- |  :----: |  :----: |  :----: |  :----: |  :----: |
> | Block 4      | 46.54/81.49      |**27.88**/92.18    | **38.26**/88.34      | 46.06/89.33      | 39.69/87.84      |
> | Block 3      | 49.61/91.42      |39.91/92.01    | 51.82/88.32      | 41.84/91.44      | 45.80/90.80      |
> | Block 3+4  | **41.31**/**91.91**      |29.27/**94.07**    | 39.34/**90.93**      | **37.29**/**91.70**      | **36.80**/**92.15**      |
>
> As can be observed above, Block 4 feature slightly outperforms Block 3 feature on SUN and Places, while Block 3 feature has an advantage on iNaturalist and Textures. This might stem from the fact that the intermediate features at different layers would focus on different semantic information. The decision cues for OOD detection are very likely to be different for Block 3 and Block 4 features. ***The fusion could leverage the distinguishable information of both features and achieves a good compromise for better performance***. We have added this explanation in the revised method section of the paper.
>
> _____

---

> ### Author Response · Authors · 2022-08-01
> **Response to Reviewer kEL5 (Part 2/2); Thanks for your encouraging feedback and constructive comments!**
>
> ___
>
> **5. How RankFeat Helps the Distribution Get Squeezed and Biased**
>
> Thanks for the insightful comment! As revealed in Fig. 2 of the paper, the score distributions of ID and OOD data are indeed more squeezed and biased (skewed). In particular, the OOD distribution becomes more skewed than the ID data.
>
> This can be understood from the analysis of our upper bound $\texttt{RankFeat}(\mathbf{x}) <\frac{1}{HW} \Big(\sum_{i=1}^{N} \mathbf{s_i}- \mathbf{s_{1}}\Big)||\mathbf{W}||_ \infty +  ||\mathbf{b}||_ \infty + \log(Q)$. The subtraction of $\mathbf{s_{1}}||\mathbf{W}||_ \infty $ would largely reduce the numerical range of both ID and OOD scores, which could squeeze the score distributions. Since the dominant singular value $\mathbf{s_{1}}$ contributes the most to the score, removing $\mathbf{s_{1}}||\mathbf{W}||_ \infty$ is likely to make many samples have similar scores. This would concentrate samples in a small region and further skew the distribution. Consider that the OOD feature tends to have a much larger $\mathbf{s_{1}}$. This would have a greater impact on the OOD data and skew the OOD score distribution more. We have added the related discussion in the theoretical analysis of the revised paper.
>
> ___
>
> Thanks again and it is more than welcome to post any further comments.

---

### Author Response · Authors · 2022-08-01
**Paper Revision and Summary of Changes**

We sincerely thank all the reviewers for their careful reviews and constructive suggestions, which indeed helps to shape the paper better. &#x1F600; &#x1F600; &#x1F600;

___


We appreciate the reviewers' common sentiment that our work is
**well-motivated** ($\color{red}{R1}$, $\color{brown}{R4}$), **novel** ($\color{red}{R1}$, $\color{green}{R3}$), **simple and intuitive** ($\color{red}{R1}$, $\color{brown}{R4}$), **written clearly and organized well** ($\color{red}{R1}$, $\color{brown}{R4}$), and has **rich experiments that show strong performance improvements and good application diversity** ($\color{green}{R3}$, $\color{brown}{R4}$). We are also glad that $\color{blue}{R2}$ thinks our method is **interesting**, and $\color{red}{R1}$ finds that our work has **a good alignment of theoretical effect and empirical results**.

The paper has been significantly revised to address the reviewers' concerns and to absorb the reviewers' suggestions. The changes have been highlighted in $\color{red}{red}$ in both the main paper and the supplementary material. We have summarized the changes as follows:

**1. More visualization of Fig. 1(a) ($\color{red}{R1}$, $\color{green}{R3}$, $\color{brown}{R4}$)**

We have added the top-5 singular value distributions of every dataset to Sec. D.1 of the revised supplementary material. The novel observation consistently holds: *the dominant singular value of OOD features always tends to be significantly larger than that of ID features on every OOD dataset* The caption of Fig. 1 is also updated with specific dataset information.

**2. Explanation of how fusion helps to improve performance ($\color{red}{R1}$)**

We discussed how the fusion improves the score to the revised method section (line95)

**3. Explanation of how RankFeat makes the score distributions squeezed and skewed ($\color{red}{R1}$)**

We add the related explanation and discussion to line134 of the revised theoretical analysis section.

**4. More evaluation on other benchmarks ($\color{blue}{R2}$, $\color{brown}{R4}$)**

The additional experiments of ***comparison against MOS and KL Matching***, ***of Species benchmark***, ***of CIFAR10 versus CIFAR100***, and ***of one-class CIFAR10 and CIFAR100*** have been added to the Sec. B of the supplementary material. We might move the experiments of Species and of comparison against MOS and KL Matching to the main paper when we are allowed for an additional content page.

**5. Clarification that our method is post hoc ($\color{blue}{R2}$)**

In the revised Sec B.3 of the supplementary material, we have clarified that our method is *post hoc*, and the implementation details are reported to describe how we obtain CIFAR models.

**6. Notation issues and how SVD is applied ($\color{green}{R3}$)**

We have corrected the notation issues throughout the paper and have explained how SVD is applied detailedly in the revised method section. The pseudo-code is also attached to Sec. A of the supplementary material.

**7. PCA-based explanation of why OOD has a larger dominant singular value ($\color{green}{R3}$)**

The PCA-based explanation has been added to Sec. E of the revised supplementary material. We also briefly discussed this point in the revised introduction to show the intuition behind our observation.

**8. Formal Theorem and Proof ($\color{green}{R3}$)**

In the revised paper, we reformulate the theoretical analysis into a more formal proposition-proof format. Also, we have added the formal and rigorous theorem of MP law as well as the proof to Sec. F of the revised supplementary.

**9. Explanation of why we use Energy score ($\color{brown}{R4}$)**

We explained why we use Energy score as the base function at line120 of the revised theoretical analysis section.

**10. Milder Theoretical Analysis and Clarification about Assumption ($\color{brown}{R4}$)**

We have revised Sec. 3 of the paper to give the impression that our bound analysis strives to give insights instead of giving strong score guarantee. Also, we have explicitly emphasized our assumption in the revised Abstract and Introduction.

___

*For brevity, we refer to reviewers $\color{red}{kEL5}$ as $\color{red}{R1}$, $\color{blue}{dq6u}$ as $\color{blue}{R2}$, $\color{green}{Ft9S}$ as $\color{green}{R3}$, and $\color{brown}{SMeo}$ as $\color{brown}{R4}$ respectively.

---

### Meta-Review · Area_Chair_S7FG · 2022-08-26

**Recommendation:** Accept
**Confidence:** Less certain

**Metareview:**

Thanks for your submission to NeurIPS.

This paper generated quite a bit of discussion, with several reviewers having lengthy discussions with the authors on various points in the paper.  At the end of the day, it seems that three of the four reviewers are mostly happy with the paper (with scores of 6, 6, 5, though the 5 reviewer indicated that they would raise their score to 6 but never did, so I'm assuming this is 6, 6, 6).  One of the reviewers was more negative, giving a 3.

The biggest issue of the negative reviewer revolves around the experimental setup, and in particular the comparison of post-hoc and non-post-hoc methods (or lack thereof) in the experimental results.  Though I'm not sure the reviewer was ever fully satisfied with the resulting experiments, I do see the differences between the experimental methodologies and am satisfied that the experiments presented in the paper are sufficient and reasonable.  Given that the other reviewers are happy with the paper (and I've also read the paper and am OK with it), I will recommend accepting the paper.

When preparing a final version of the manuscript, please do keep in mind the many comments of the reviewers, and try to address these as much as possible.

**Award:**

No

---

### Decision · Program_Chairs · 2022-09-14

Accept